# Global Optimality in Bivariate Gradient-based DAG Learning

**Chang Deng**[†*]    **Kevin Bello**[†,‡]    **Pradeep Ravikumar**[‡]    **Bryon Aragam**[†]

[†]Booth School of Business, University of Chicago, Chicago, IL 60637
[‡]Machine Learning Department, Carnegie Mellon University, Pittsburgh, PA 15213

## Abstract

Recently, a new class of non-convex optimization problems motivated by the statistical problem of learning an acyclic directed graphical model from data has attracted significant interest. While existing work uses standard first-order optimization schemes to solve this problem, proving the global optimality of such approaches has proven elusive. The difficulty lies in the fact that unlike other non-convex problems in the literature, this problem is not "benign", and possesses multiple spurious solutions that standard approaches can easily get trapped in. In this paper, we prove that a simple path-following optimization scheme globally converges to the global minimum of the population loss in the bivariate setting.

## 1 Introduction

Over the past decade, non-convex optimization has become a major topic of research within the machine learning community, in part due to the successes of training large-scale models with simple first-order methods such as gradient descent—along with their stochastic and accelerated variants—in spite of the non-convexity of the loss function. A large part of this research has focused on characterizing which problems have *benign* loss landscapes that are amenable to the use of gradient-based methods, i.e., there are no spurious local minima, or they can be easily avoided. By now, several theoretical results have shown this property for different non-convex problems such as: learning a two hidden unit ReLU network [48], learning (deep) over-parameterized quadratic neural networks [43, 27], low-rank matrix recovery [19, 13, 3], learning a two-layer ReLU network with a single non-overlapping convolutional filter [6], semidefinite matrix completion [4, 20], learning neural networks for binary classification with the addition of a single special neuron [30], and learning deep networks with independent ReLU activations [26, 11], to name a few.

Recently, a new class of non-convex optimization problems due to Zheng et al. [51] have emerged in the context of learning the underlying structure of a structural equation model (SEM) or Bayesian network. This underlying structure is typically represented by a directed acyclic graph (DAG), which makes the learning task highly complex due to its combinatorial nature. In general, learning DAGs is well-known to be NP-complete [8, 10]. The key innovation in Zheng et al. [51] was the introduction of a differentiable function $h$, whose level set at zero *exactly* characterizes DAGs. Thus, replacing the challenges of combinatorial optimization by those of non-convex optimization. Mathematically, this class of non-convex problems take the following general form:

$$\min_{\Theta} f(\Theta) \text{ subject to } h(W(\Theta)) = 0, \tag{1}$$

where $\Theta \in \mathbb{R}^l$ represents the model parameters, $f : \mathbb{R}^l \to \mathbb{R}$ is a (possibly non-convex) smooth loss function (sometimes called a *score function*) that measures the fitness of $\Theta$, and $h : \mathbb{R}^{d \times d} \to [0, \infty)$

---

[*]changdeng@chicagobooth.edu

is a smooth **non-convex** function that takes the value of zero if and only if the induced weighted adjacency matrix of $d$ nodes, $W(\Theta)$, corresponds to a DAG.

Given the smoothness of $f$ and $h$, problem (1) can be solved using off-the-shelf nonlinear solvers, which has driven a series of remarkable developments in structure learning for DAGs. Multiple empirical studies have demonstrated that global or near-global minimizers for (1) can often be found in a variety of settings, such as linear models with Gaussian and non-Gaussian noises [e.g., 51, 34, 1], and non-linear models, represented by neural networks, with additive Gaussian noises [e.g., 29, 52, 49, 1]. The empirical success for learning DAGs via (1), which started with the NOTEARS method of Zheng et al. [51], bears a resemblance to the advancements in deep learning, where breakthroughs like AlexNet significantly boosted the field's recognition, even though there were notable successes before it.

Importantly, the reader should note that the majority of applications in ML consist of solving a *single unconstrained* non-convex problem. In contrast, the class of problems (1) contains a non-convex constraint. Thus, researchers have considered some type of penalty method such as the augmented Lagrangian [51, 52], quadratic penalty [35], and a log-barrier [1]. In all cases, the penalty approach consists in solving a *sequence* of unconstrained non-convex problems, where the constraint is enforced progressively [see e.g. 2, for background]. In this work, we will consider the following form of penalty:

$$\min_{\Theta} g_{\mu_k}(\Theta) := \mu_k f(\Theta) + h(W(\Theta)). \tag{2}$$

It was shown by Bello et al. [1] that due to the invexity property of $h$,[2] solutions to (2) will converge to a DAG as $\mu_k \to 0$. However, no guarantees on local/global optimality were given.

With the above considerations in hand, one is inevitably led to ask the following questions:

(i) *Are the loss landscapes $g_{\mu_k}(\Theta)$ benign for different $\mu_k$?*

(ii) *Is there a (tractable) solution path $\{\Theta_k\}$ that converges to a global minimum of* (1)*?*

Due to the NP-completeness of learning DAGs, one would expect the answer to (i) to be negative in its most general form. Moreover, it is known from the classical theory of constrained optimization [e.g. 2] that if we can *exactly* and *globally* optimize (1) for each $\mu_k$, then the answer to (ii) is affirmative. This is not a practical algorithm, however, since the problem (1) is nonconvex. Thus we seek a solution path that can be tractably computed in practice, e.g. by gradient descent.

In this work, we focus on perhaps the simplest setting where interesting phenomena take place. That is, a linear SEM with two nodes (i.e., $d = 2$), $f$ is the population least squared loss (i.e., $f$ is convex), and $\Theta_k$ is defined via gradient flow with warm starts. More specifically, we consider the case where $\Theta_k$ is obtained by following the gradient flow of $g_{\mu_k}$ with initial condition $\Theta_{k-1}$.

Under this setting, to answer (i), it is easy to see that for a large enough $\mu_k$, the convex function $f$ dominates and we can expect a benign landscape, i.e., a (almost) convex landscape. Similarly, when $\mu_k$ approaches zero, the invexity of $h$ kicks in and we could expect that all stationary points are (near) global minimizers.[3] That is, at the extremes $\mu_k \to \infty$ and $\mu_k \to 0$, the landscapes seem well-behaved, and the reader might wonder if it follows that for any $\mu_k \in [0, \infty)$ the landscape is well-behaved. We answer the latter in the *negative* and show that there always exists a $\tau > 0$ where the landscape of $g_{\mu_k}$ is non-benign for any $\mu_k < \tau$, namely, there exist three stationary points: i) A saddle point, ii) A spurious local minimum, and iii) The global minimum. In addition, each of these stationary points have wide basins of attractions, thus making the initialization of the gradient flow for $g_{\mu_k}$ crucial. Finally, we answer (ii) in the affirmative and provide an explicit scheduling for $\mu_k$ that guarantees the asymptotic convergence of $\Theta_k$ to the global minimum of (1). Moreover, we show that this scheduling cannot be arbitrary as there exists a sequence of $\{\mu_k\}$ that leads $\{\Theta_k\}$ to a spurious local minimum.

Overall, we establish the first set of results that study the optimization landscape and global optimality for the class of problems (1). We believe that this comprehensive analysis in the bivariate case provides a valuable starting point for future research in more complex settings.

---

[2]An invex function is any function where all its stationary points are global minima. It is worth noting that the composite objective in (2) is not necessarily invex, even when $f$ is convex.

[3]This transition or path, from an optimizer of a simple function to an optimizer of a function that closely resembles the original constrained formulation, is also known as a *homotopy*.

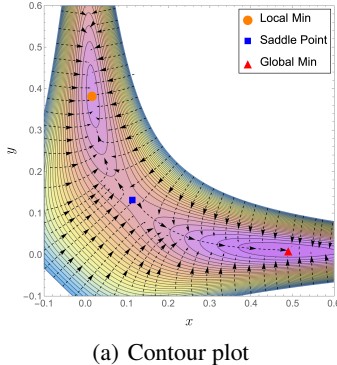

(a) Contour plot

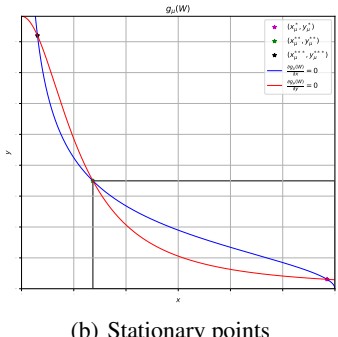

(b) Stationary points

Figure 1: Visualizing the nonconvex landscape. (a) A contour plot of $g_\mu$ for $a = 0.5$ and $\mu = 0.005$ (see Section 2 for definitions). We only show a section of the landscape for better visualization. The solid lines represent the contours, while the dashed lines represent the vector field $-\nabla g_\mu$. (b) Stationary points of $g_\mu$, $r(y; \mu) = 0$ and $r(x; \mu) = 0$ (see Section 4 for definitions).

**Remark 1.** *We emphasize that solving* (1) *in the bivariate case is not an inherently difficult problem. Indeed, when there are only two nodes, there are only two DAGs to distinguish and one can simply fit $f$ under the only two possible DAGs, and select the model with the lowest value for $f$. However, **evaluating $f$ for each possible DAG structure clearly cannot scale beyond 10 or 20 nodes, and is not a standard algorithm for solving** (1). Instead, here **our focus is on studying how** (1) **is actually being solved in practice**, namely, by solving unconstrained non-convex problems in the form of* (2). *Previous work suggests that such gradient-based approaches indeed scale well to hundreds and even thousands of nodes [e.g. 51, 34, 1].*

## 1.1 Our Contributions

More specifically, we make the following contributions:

1. We present a homotopy-based optimization scheme (Algorithm 2) to find global minimizers of the program (1) by iteratively decreasing the penalty coefficient according to a given schedule. Gradient flow is used to find the stationary points of (2) at each step, starting from the previous solution.

2. We prove that Algorithm 2 converges *globally* (i.e. regardless of initialization for $W$) to the *global* minimum (Theorem 1).

3. We show that the non-convex program (1) is indeed non-benign, and naïve implementation of black-box solvers are likely to get trapped in a bad local minimum. See Figure 1 (a).

4. Experimental results verify our theory, consistently recovering the global minimum of (1), regardless of initialization or initial penalty value. We show that our algorithm converges to the global minimum while naïve approaches can get stuck.

The analysis consists of three main parts: First, we explicitly characterize the trajectory of the stationary points of (2). Second, we classify the number and type of all stationary points (Lemma 1) and use this to isolate the desired global minimum. Finally, we apply Lyapunov analysis to identify the basin of attraction for each stationary point, which suggests a schedule for the penalty coefficient that ensures that the gradient flow is initialized within that basin at the previous solution.

## 1.2 Related Work

The class of problems (1) falls under the umbrella of score-based methods, where given a score function $f$, the goal is to identify the DAG structure with the lowest score possible [9, 22]. We shall note that learning DAGs is a very popular structure model in a wide range of domains such as biology [40], genetics [50], and causal inference [44, 39], to name a few.

**Score-based methods that consider the combinatorial constraint.** Given the ample set of score-based methods in the literature, we briefly mention some classical works that attempt to optimize $f$ by considering the combinatorial DAG constraint. In particular, we have approximate algorithms such as the greedy search method of Chickering et al. [10], order search methods [45, 41, 38], the LP-relaxation method of Jaakkola et al. [24], and the dynamic programming approach of Loh and Bühlmann [31]. There are also exact methods such as GOBNILP [12] and Bene [42], however, these algorithms only scale up to $\approx 30$ nodes.

**Score-based methods that consider the continuous non-convex constraint $h$.** The following works are the closest to ours since they attempt to solve a problem in the form of (1). Most of these developments either consider optimizing different score functions $f$ such as ordinary least squares [51, 52], the log-likelihood [29, 34], the evidence lower bound [49], a regret function [53]; or consider different differentiable characterizations of acyclicity $h$ [49, 1]. However, none of the aforementioned works provide any type of optimality guarantee. Few studies have examined the optimization intricacies of problem (1). Wei et al. [47] investigated the optimality issues and provided *local* optimality guarantees under the assumption of convexity in the score $f$ and linear models. On the other hand, Ng et al. [35] analyzed the convergence to (local) DAGs of generic methods for solving nonlinear constrained problems, such as the augmented Lagrangian and quadratic penalty methods. In contrast to both, our work is the first to study global optimality and the loss landscapes of actual methods used in practice for solving (1).

**Bivariate causal discovery.** Even though in a two-node model the discrete DAG constraint does not pose a major challenge, the bivariate setting has been subject to major research in the area of causal discovery. See for instance [36, 16, 32, 25] and references therein.

**Penalty and homotopy methods.** There exist classical global optimality guarantees for the penalty method if $f$ and $h$ were convex functions, see for instance [2, 5, 37]. However, to our knowledge, there are no global optimality guarantees for general classes of non-convex constrained problems, let alone for the specific type of non-convex functions $h$ considered in this work. On the other hand, homotopy methods (also referred to as continuation or embedding methods) are in many cases capable of finding better solutions than standard first-order methods for non-convex problems, albeit they typically do not come with global optimality guarantees either. When homotopy methods come with global optimality guarantees, they are commonly computationally more intensive as it involves discarding solutions, thus, closely resembling simulated annealing methods, see for instance [15]. Authors in [21] characterize a family of non-convex functions where a homotopy algorithm provably converges to a global optimum. However, the conditions for such family of non-convex functions are difficult to verify and are very restrictive; moreover, their homotopy algorithm involves Gaussian smoothing, making it also computationally more intensive than the procedure we study here. Other examples of homotopy methods in machine learning include [7, 18, 46, 17, 23], in all these cases, no global optimality guarantees are given.

## 2 Preliminaries

The objective $f$ we consider can be easily written down as follows:

$$f(W) = \frac{1}{2}\mathbb{E}_X \left[ \|X - W^\top X\|_2^2 \right], \tag{3}$$

where $X \in \mathbb{R}^2$ is a random vector and $W \in \mathbb{R}^{2\times 2}$. Although not strictly necessary for the developments that follow, we begin by introducing the necessary background on linear SEM that leads to this objective and the resulting optimization problem of interest.

**The bivariate model.** Let $X = (X_1, X_2) \in \mathbb{R}^2$ denote the random variables in the model, and let $N = (N_1, N_2) \in \mathbb{R}^2$ denote a vector of independent errors. Then a linear SEM over $X$ is defined as $X = W_*^\top X + N$, where $W_* \in \mathbb{R}^{2\times 2}$ is a weighted adjacency matrix encoding the coefficients in the linear model. In order to represent a valid Bayesian network for $X$ [see e.g. 39, 44, for details], the matrix $W_*$ must be acyclic: More formally, the weighted graph induced by the adjacency matrix $W_*$ must be a DAG. This (non-convex) acyclicity constraint represents the major computational hurdle that must overcome in practice (cf. Remark 1).

The goal is to recover the matrix $W_*$ from the random vector $X$. Since $W_*$ is acyclic, we can assume the diagonal of $W_*$ is zero (i.e. no self-loops). Thus, under the bivariate linear model, it then suffices to consider two parameters $x$ and $y$ that define the matrix of parameters[4]

$$W = W(x, y) = \begin{pmatrix} 0 & x \\ y & 0 \end{pmatrix} \tag{4}$$

For notational simplicity, we will use $f(W)$ and $f(x, y)$ interchangeably, similarly for $h(W)$ and $h(x, y)$. Without loss of generality, we write the underlying parameter as

$$W_* = \begin{pmatrix} 0 & a \\ 0 & 0 \end{pmatrix} \tag{5}$$

which implies

$$X = W_*^\top X + N \implies \begin{cases} X_1 = N_1, \\ X_2 = aX_1 + N_2. \end{cases}$$

In general, we only require $N_i$ to have finite mean and variance, hence we *do not* assume Gaussianity. We assume that $\mathrm{Var}[N_1] = \mathrm{Var}[N_2]$, and for simplicity, we consider $\mathbb{E}[N] = 0$ and $\mathrm{Cov}[N] = I$, where $I$ denotes the identity matrix. Finally, in the sequel we assume w.l.o.g. that $a > 0$.

**The population least squares.**   In this work, we consider the population squared loss defined by (3). If we equivalently write $f$ in terms of $x$ and $y$, then we have: $f(W) = ((1-ay)^2 + y^2 + (a-x)^2 + 1)/2$. In fact, the population loss can be substituted with empirical loss. In such a case, our algorithm can still attain the global minimum, $W_G$, of problem (6). However, the output $W_G$ will serve as an empirical estimation of $W_*$. An in-depth discussion on this topic can be found in Appendix B

**The non-convex function $h$.**   We use the continuous acyclicity characterization of Yu et al. [49], i.e., $h(W) = \mathrm{Tr}((I + \frac{1}{d}W \circ W)^d) - d$, where $\circ$ denotes the Hadamard product. Then, for the bivariate case, we have $h(W) = x^2y^2/2$. We note that the analysis presented in this work is not tailored to this version of $h$, that is, we can use the same techniques used throughout this work for other existing formulations of $h$, such as the trace of the matrix exponential [51], and the log-det formulation [1]. Nonetheless, here we consider that the polynomial formulation of Yu et al. [49] is more amenable for the analysis.

**Remark 2.** *Our restriction to the bivariate case highlights the simplest setting in which this problem exhibits nontrivial behaviour. Extending our analysis to higher dimensions remains a challenging future direction, however, we emphasize that even in two-dimensions this problem is nontrivial. Our approach is similar to that taken in other parts of the literature that started with simple cases (e.g. single-neuron models in deep learning).*

**Remark 3.** *It is worth noting that our choice of the population least squares is not arbitrary. Indeed, for linear models with identity error covariance, such as the model considered in this work, it is known that the global minimizer of the population squared loss is unique and corresponds to the underlying matrix $W_*$. See Theorem 7 in [31].*

Gluing all the pieces together, we arrive to the following version of (1) for the bivariate case:

$$\min_{x,y} f(x, y) := \frac{1}{2}((1 - ay)^2 + y^2 + (a - x)^2 + 1) \text{ subject to } h(x, y) := \frac{x^2y^2}{2} = 0. \tag{6}$$

Moreover, for any $\mu \geq 0$, we have the corresponding version of (2) expressed as:

$$\min_{x,y} g_\mu(x, y) := \mu f(x, y) + h(x, y) = \frac{\mu}{2}((1 - ay)^2 + y^2 + (a - x)^2 + 1) + \frac{x^2y^2}{2}. \tag{7}$$

To conclude this section, we present a visualization of the landscape of $g_\mu(x, y)$ in Figure 1 (a), for $a = 0.5$ and $\mu = 0.005$. We can clearly observe the non-benign landscape of $g_\mu$, i.e., there exists a spurious local minimum, a saddle point, and the global minimum. In particular, we can see that the basin of attraction of the spurious local minimum is comparable to that of the global minimum, which is problematic for a local algorithm such as the gradient flow (or gradient descent) as it can easily get trapped in a local minimum if initialized in the wrong basin.

---

[4]Following the notation in (1), for the bivariate model we simply have $\Theta \equiv (x, y)$ and $W(\Theta) \equiv \begin{pmatrix} 0 & x \\ y & 0 \end{pmatrix}$. Although $x$ is used to represent edge weights and not data as in (3), this distinction should not lead to confusion

**Algorithm 1:** GradientFlow($f, z_0$)

---

1: set $z(0) = z_0$
2: $\frac{d}{dt} z(t) = -\nabla f(z(t))$
3: **return** $\lim_{t \to \infty} z(t)$

---

**Algorithm 2:** Homotopy algorithm for solving (1).

---

**Input:** Initial $W_0 = W(x_0, y_0)$, $\mu_0 \in \left[ \frac{a^2}{4(a^2+1)^3}, \frac{a^2}{4} \right)$

**Output:** $\{W_{\mu_k}\}_{k=0}^{\infty}$

1 $W_{\mu_0} \leftarrow \texttt{GradientFlow}(g_{\mu_0}, W_0)$
2 **for** $k = 1, 2, \ldots$ **do**
3 $\quad$ Let $\mu_k = (2/a)^{2/3} \mu_{k-1}^{4/3}$
4 $\quad$ $W_{\mu_k} \leftarrow \texttt{GradientFlow}(g_{\mu_k}, W_{\mu_{k-1}})$
5 **end**

---

## 3 A Homotopy-Based Approach and Its Convergence to the Global Optimum

To fix notation, let us write $W_k := W_{\mu_k} := \left( \begin{smallmatrix} 0 & x_{\mu_k} \\ y_{\mu_k} & 0 \end{smallmatrix} \right)$. and let $W_{\mathsf{G}}$ denote the global minimizer of (6). In this section, we present our main result, which provides conditions under which solving a series of unconstrained problems (7) with first-order methods will converge to the global optimum $W_{\mathsf{G}}$ of (6), in spite of facing non-benign landscapes. Recall that from Remark 3, we have that $W_{\mathsf{G}} = \left( \begin{smallmatrix} 0 & a \\ 0 & 0 \end{smallmatrix} \right)$. Since we use gradient flow path to connect $W_{\mu_k}$ and $W_{\mu_{k+1}}$, we specify this path in Procedure 1 for clarity. Although the theory here assumes continuous-time gradient flow with $t \to \infty$, see Section 5 for an iteration complexity analysis for (discrete-time) gradient descent, which is a straightforward consequence of the continuous-time theory.

In Algorithm 2, we provide an explicit regime of initialization for the homotopy parameter $\mu_0$ and a specific scheduling for $\mu_k$ such that the solution path found by Algorithm 2 will converge to the global optimum of (6). This is formally stated in Theorem 1, whose proof is given in Section 5.

**Theorem 1.** *For any initialization $W_0$ and $a \in \mathbb{R}$, the solution path provided in Algorithm 2 converges to the global optimum of* (6)*, i.e.,*

$$\lim_{k \to \infty} W_{\mu_k} = W_{\mathsf{G}}.$$

A few observations regarding Algorithm 2: Observe that when the underlying model parameter $a \gg 0$, the regime of initialization for $\mu_0$ is wider; on the other hand, if $a$ is closer to zero then the interval for $\mu_0$ is much narrower. As a concrete example, if $a = 2$ then it suffices to have $\mu_0 \in [0.008, 1)$; whereas if $a = 0.1$ then the regime is about $\mu_0 \in [0.0089, 0.01)$. This matches the intuition that for a "stronger" value of $a$ it should be easier to detect the right direction of the underlying model. Second, although in Line 3 we set $\mu_k$ in a specific manner, it actually suffices to have

$$\mu_k \in \left[ (\frac{\mu_{k-1}}{2})^{2/3}(a^{1/3} - \sqrt{a^{2/3} - (4\mu_{k-1})^{1/3}})^2, \mu_{k-1} \right).$$

We simply chose a particular expression from this interval for clarity of presentation; see the proof in Section 5 for details.

As presented, Algorithm 2 is of theoretical nature in the sense that the initialization for $\mu_0$ and the decay rate for $\mu_k$ in Line 3 depend on the underlying parameter $a$, which in practice is unknown. In Algorithm 3, we present a modification that is independent of $a$ and $W_*$. By assuming instead a lower bound on $a$, which is a standard assumption in the literature, we can prove that Algorithm 3 also converges to the global minimum:

**Corollary 1.** *Initialize $\mu_0 = \frac{1}{27}$. If $a > \sqrt{5/27}$ then for any initialization $W_0$, Algorithm 3 outputs the global optimal solution to* (6)*, i.e.*

$$\lim_{k \to \infty} W_{\mu_k} = W_{\mathsf{G}}.$$

For more details on this modification, see Appendix A.

**Algorithm 3:** Practical (i.e. independent of $a$ and $W_*$) homotopy algorithm for solving (1).

---

**Input:** Initial $W_0 = W(x_0, y_0)$
**Output:** $\{W_{\mu_k}\}_{k=0}^{\infty}$

1   $\mu_0 \leftarrow 1/27$
2   $W_{\mu_0} \leftarrow \texttt{GradientFlow}(g_{\mu_0}, W_0)$
3   **for** $k = 1, 2, \ldots$ **do**
4      Let $\mu_k = \left(2/\sqrt{5\mu_0}\right)^{2/3} \mu_{k-1}^{4/3}$
5      $W_{\mu_k} \leftarrow \texttt{GradientFlow}(g_{\mu_k}, W_{\mu_{k-1}})$
6   **end**

---

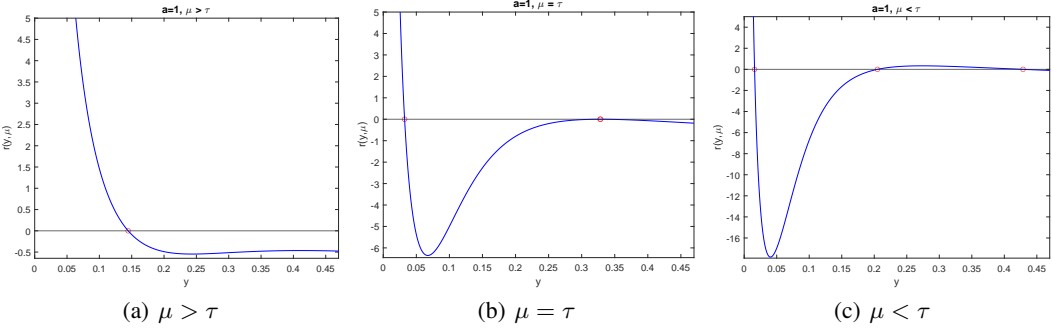

(a) $\mu > \tau$         (b) $\mu = \tau$         (c) $\mu < \tau$

Figure 2: The behavior of $r(y; \mu)$ for different $\mu$. Here, for $\mu > \tau$, there exists a single solution to $r(y; \mu) = 0$, which implies there is one stationary point in Equation (7). When $\mu = \tau$, two solutions are found for $r(y; \mu) = 0$, suggesting that there are two stationary points in Equation (7). Conversely, when $\mu < \tau$, we observe three solutions for $r(y; \mu) = 0$, indicating that there are three stationary points in Equation (7)- a local optimum, a saddle point, and a global optimum.

## 4   A Detailed Analysis of the Evolution of the Stationary Points

The homotopy approach in Algorithm 2 relies heavily on how the stationary points of (7) behave with respect to $\mu_k$. In this section, we dive deep into the properties of these critical points.

By analyzing the first-order conditions for $g_\mu$, we first narrow our attention to the region $A = \{0 \leq x \leq a, 0 \leq y \leq \frac{a}{a^2+1}\}$. By solving the resulting equations, we obtain an equation that only involves the variable $y$:

$$r(y; \mu) = \frac{a}{y} - \frac{\mu a^2}{(y^2 + \mu)^2} - (a^2 + 1). \tag{8}$$

Likewise, we can find an equation only involving the variable $x$:

$$t(x; \mu) = \frac{a}{x} - \frac{\mu a^2}{(\mu(a^2 + 1) + x^2)^2} - 1. \tag{9}$$

To understand the behavior of the stationary points of $g_\mu(W)$, we can examine the characteristics of $t(x; \mu)$ in the range $x \in [0, a]$ and the properties of $r(y; \mu)$ in the interval $y \in [0, \frac{a}{a^2+1}]$.

In Figures 2 and 3, we show the behavior of $r(y; \mu)$ and $t(x; \mu)$ for $a = 1$. Theorems 5 and 6 in the appendix establish the existence of a $\tau > 0$ with the following useful property:

**Corollary 2.** *There exists $\mu < \tau$ such that the equation $\nabla g_\mu(W) = 0$ has three different solutions, denoted as $W_\mu^*, W_\mu^{**}, W_\mu^{***}$. Then,*

$$\lim_{\mu \to 0} W_\mu^* = \begin{bmatrix} 0 & a \\ 0 & 0 \end{bmatrix}, \ \lim_{\mu \to 0} W_\mu^{**} = \begin{bmatrix} 0 & 0 \\ 0 & 0 \end{bmatrix}, \ \lim_{\mu \to 0} W_\mu^{***} = \begin{bmatrix} 0 & 0 \\ \frac{a}{a^2+1} & 0 \end{bmatrix}$$

Note that the interesting regime takes place when $\mu < \tau$. Then, we characterize the stationary points as either local minima or saddle points:

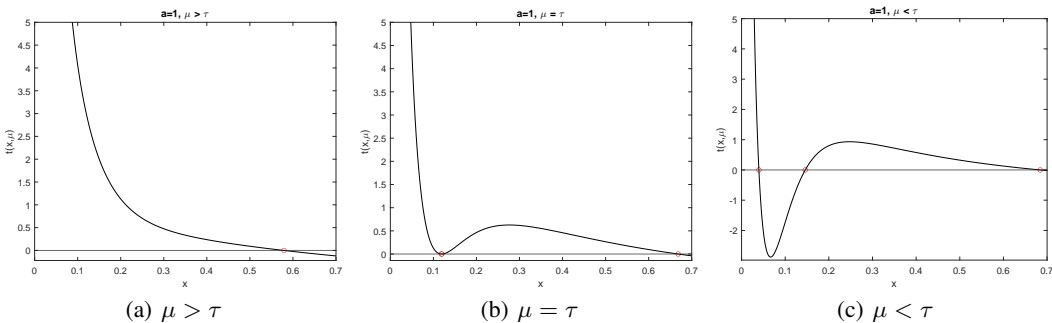

Figure 3: The behavior of $t(x; \mu)$ for different $\mu$. Here, for $\mu > \tau$, there exists a single solution to $t(x; \mu) = 0$, which implies there is one stationary point in Equation (7). When $\mu = \tau$, two solutions are found for $t(x; \mu) = 0$, suggesting that there are two stationary points in Equation (7). Conversely, when $\mu < \tau$, we observe three solutions for $t(x; \mu) = 0$, indicating that there are three stationary points in Equation (7)- a local optimum, a saddle point, and a global optimum.

**Lemma 1.** *Let $\mu < \tau$, then $g_\mu(W)$ has two local minima at $W_\mu^*, W_\mu^{***}$, and a saddle point at $W_\mu^{**}$.*

With the above results, it has been established that $W_\mu^*$ converges to the global minimum $W_G$ as $\mu \to 0$. In the following section for the proof of Theorem 1, we perform a thorough analysis on how to track $W_\mu^*$ and avoid the local minimum at $W_\mu^{**}$ by carefully designing the scheduling for $\mu_k$.

## 5 Convergence Analysis: From continuous to discrete

We now discuss the iteration complexity of our method when gradient descent is used in place of gradient flow. We begin with some preliminaries regarding the continuous-time analysis.

### 5.1 Continuous case: Gradient flow

The key to ensuring the convergence of gradient flow to $W_\mu^*$ is to accurately identify the basin of attraction of $W_\mu^*$. The following lemma provides a region that lies within such basin of attraction.

**Lemma 2.** *Define $B_\mu = \{(x, y) \mid x_\mu^{**} < x \leq a, 0 \leq y < y_\mu^{**}\}$. Run Algorithm 1 with input $f = g_\mu(x, y), z_0 = (x(0), y(0))$ where $(x(0), y(0)) \in B_\mu$, then $\forall t \geq 0$, we have that $(x(t), y(t)) \in B_\mu$ and $\lim_{t \to \infty}(x(t), y(t)) = (x_\mu^*, y_\mu^*)$.*

In Figure 1 (b), the lower-right rectangle corresponds to $B_\mu$. Lemma 2 implies that the gradient flow with any initialization inside $B_{\mu_{k+1}}$ will converge to $W_{\mu_{k+1}}^*$ at last. Then, by utilizing the previous solution $W_{\mu_k}^*$ as the initial point, as long as it lies within region $B_{\mu_{k+1}}$, the gradient flow can converge to $W_{\mu_{k+1}}^*$, thereby achieving the goal of tracking $W_{\mu_{k+1}}^*$. Following the scheduling for $\mu_k$ prescribed in Algorithm 2 provides a sufficient condition to ensure that will happen.

The following lemma, with proof in the appendix, is used for the Proof of Theorem 1. It provides a lower bound for $y_\mu^{**}$ and upper bound for $y_\mu^*$.

**Lemma 3.** *If $\mu < \tau$, then $y_\mu^{**} > \sqrt{\mu}$, and $\frac{(4\mu)^{1/3}}{2}\left(a^{1/3} - \sqrt{a^{2/3} - (4\mu)^{1/3}}\right) > y_\mu^*$.*

**Proof of Theorem 1.** Consider that we are at iteration $k + 1$ of Algorithm 2, then $\mu_{k+1} < \mu_k$. If $\mu_k > \tau$ and $\mu_{k+1} > \tau$, then there is only one stationary point for $g_{\mu_k}(x, y)$ and $g_{\mu_{k+1}}(x, y)$, thus, 1 will converge to such stationary point. Hence, let us assume $\mu_{k+1} \leq \tau$. From Theorem 6 in the appendix, we know that $x_{\mu_{k+1}}^{**} < x_{\mu_k}^*$. Then, the following relations hold:

$$y_{\mu_{k+1}}^{**} \overset{(1)}{>} \sqrt{\mu_{k+1}} \geq 2\left(\frac{\mu_k^2}{4a}\right)^{1/3} \overset{(2)}{\geq} \frac{(4\mu_k)^{1/3}}{2}\left(a^{1/3} - \sqrt{a^{2/3} - (4\mu_k)^{1/3}}\right) \overset{(3)}{>} y_{\mu_k}^*$$

Here (1) and (3) are due to Lemma 3, and (2) follows from $\sqrt{1 - x} \geq 1 - x$ for $0 \leq x \leq 1$. Then it implies that $(x_{\mu_k}^*, y_{\mu_k}^*)$ is in the region $\{(x, y) \mid x_{\mu_{k+1}}^{**} < x \leq a, 0 \leq y < y_{\mu_{k+1}}^{**}\}$. By Lemma 2, the

1 procedure will converge to $(x^*_{\mu_{k+1}}, y^*_{\mu_{k+1}})$. Finally, from Theorems 5 and 6, if $\lim_{k\to\infty} \mu_k = 0$, then $\lim_{k\to\infty} x^*_{\mu_k} = a$, $\lim_{k\to\infty} y^*_{\mu_k} = 0$, thus, converging to the global optimum, i.e.,

$$\lim_{k\to\infty} W_{\mu_k} = W_{\mathsf{G}}.$$

## 5.2 Discrete case: Gradient Descent

In Algorithms 2 and 4, gradient flow is employed to locate the next stationary points, which is not practically feasible. A viable alternative is to execute Algorithm 2, replacing the gradient flow with gradient descent. Now, at every iteration $k$, Algorithm 6 uses gradient descent to output $W_{\mu_k, \epsilon_k}$, a $\epsilon_k$ stationary point of $g_{\mu_k}$, initialized at $W_{\mu_{k-1}, \epsilon_{k-1}}$, and a step size of $\eta_k = 1/(\mu_k(a^2 + 1) + 3a^2)$. The tolerance parameter $\epsilon_k$ can significantly influence the behavior of the algorithm and must be controlled for different iterations. A convergence guarantee is established via a simplified theorem presented here. A more formal version of the theorem and a comprehensive description of the algorithm (i.e., Algorithm 6) can be found in Appendix C.

**Theorem 2** (Informal). *For any $\varepsilon_{\text{dist}} > 0$, set $\mu_0$ satisfy a mild condition, and use updating rule $\epsilon_k = \min\{\beta a \mu_k, \mu_k^{3/2}\}$, $\mu_{k+1} = (2\mu_k^2)^{2/3} \frac{(a+\epsilon_k/\mu_k)^{2/3}}{(a-\epsilon_k/\mu_k)^{4/3}}$, and let $K \equiv K(\mu_0, a, \varepsilon_{\text{dist}}) \in O\left(\ln \frac{\mu_0}{a \varepsilon_{\text{dist}}}\right)$. Then, for any initialization $W_0$, following the updated procedure above for $k = 0, \dots, K$, we have:*

$$\|W_{\mu_k, \epsilon_k} - W_{\mathsf{G}}\|_2 \le \varepsilon_{\text{dist}}$$

*that is, $W_{\mu_k, \epsilon_k}$ is $\varepsilon_{\text{dist}}$-close in Frobenius norm to global optimum $W_{\mathsf{G}}$. Moreover, the total number of gradient descent steps is upper bounded by $O\left(\left(\mu_0 a^2 + a^2 + \mu_0\right)\left(\frac{1}{a^6} + \frac{1}{\varepsilon_{\text{dist}}^6}\right)\right)$.*

# 6 Experiments

We conducted experiments to verify that Algorithms 2 and 4 both converge to the global minimum of (7). Our purpose is to illustrate two main points: First, we compare our updating scheme as given in Line 3 of Algorithm 2 against a faster-decreasing updating scheme for $\mu_k$. In Figure 4 we illustrate how a naive faster decrease of $\mu$ can lead to spurious a local minimum. Second, in Figure 5, we show that regardless of the initialization, Algorithms 2 and 4 always return the global minimum. In the supplementary material, we provide additional experiments where the gradient flow is replaced with gradient descent. For more details, please refer to Appendix F.

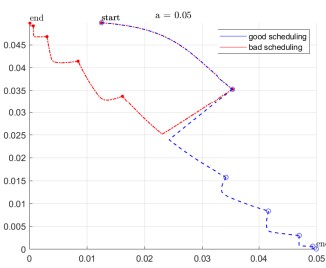

Figure 4: Trajectory of the gradient flow path for two different update rules for $\mu_k$ with same initialization and $\mu_0$. Here, "good scheduling" uses Line 3 of Algorithm 2, while "bad scheduling" uses a faster decreasing scheme for $\mu_k$ which leads the path to a spurious local minimum.

# 7 Acknowledgments and Disclosure of Funding

K. B. was supported by NSF under Grant # 2127309 to the Computing Research Association for the CIFellows 2021 Project. B.A. was supported by NSF IIS-1956330, NIH R01GM140467, and the Robert H. Topel Faculty Research Fund at the University of Chicago Booth School of Business. This work was done in part while B.A. was visiting the Simons Institute for the Theory of Computing. P.R. was supported by ONR via N000141812861, and NSF via IIS-1909816, IIS-1955532, IIS-2211907. We are also grateful for the support of the University of Chicago Research Computing Center for assistance with the calculations carried out in this work.

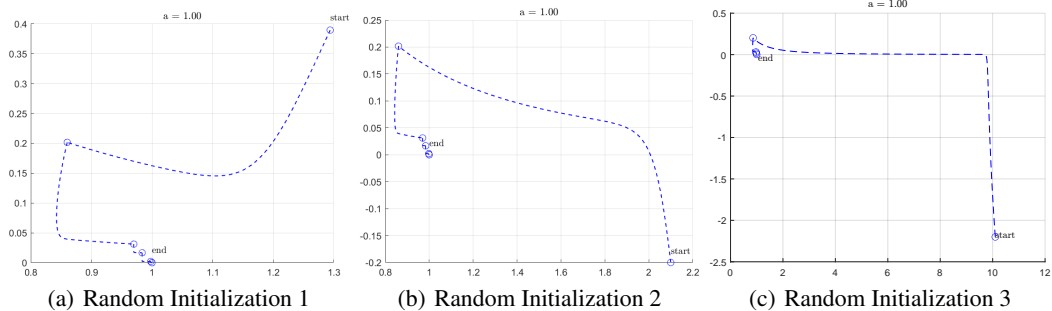

Figure 5: Trajectory of the gradient flow path with the different initializations. We observe that under a proper scheduling for $\mu_k$, they all converge to the global minimum.

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

---

**Algorithm 4:** Find a path $\{W_{\mu_k}\}$ via a particular scheduling for $\mu_k$ when $a$ is unknown.

---

**Input:** $\mu_0 \in \left[\frac{a^2}{4(a^2+1)^3}, \frac{a^2}{4}\right), \varepsilon > 0$

**Output:** $\{W_{\mu_k}\}_{k=0}^{\infty}$

1    $\widehat{a} \leftarrow \sqrt{4(\mu_0 + \varepsilon)}$                                          `// ` $\forall \varepsilon \geq 0$ `s.t.` $\widehat{a} < a$

2    $W_{\mu_0} \leftarrow \texttt{GradientFlow}(g_{\mu_0}, \mathbf{0})$

3 **for** $k = 1, 2, \ldots$ **do**

4      Let $\mu_{k+1} \in \left[(2/\widehat{a})^{2/3} \mu_k^{4/3}, \mu_k\right)$

5      $W_{\mu_{k+1}} \leftarrow \texttt{GradientFlow}(g_{\mu_{k+1}}, W_{\mu_k})$

6 **end**

7 **return** $\{W_{\mu_k}\}_{k=0}^{\infty}$

---

## A   Practical Implementation of Algorithm 2

We present a practical implementation of our homotopy algorithm in Algorithm 4. The updating scheme for $\mu_k$ is now independent of the parameter $a$, but as presented, the initialization for $\mu_0$ still depends on $a$. This is for the following reason: It is possible to make the updating scheme independent of $a$ *without imposing any additional assumptions on $a$*, as evidenced by Lemma 4 below. The initialization for $\mu_0$, however, is trickier, and we must consider two separate cases:

1. *No assumptions on $a$.* In this case, if $a$ is too small, then the problem becomes harder and the initial choice of $\mu_0$ matters.

2. *Lower bound on $a$.* If we are willing to accept a lower bound on $a$, then there is an initialization for $\mu_0$ that does not depend on $a$.

In Corollary 1, we illustrate this last point with the additional condition that $a > \sqrt{5/27}$. This essentially amounts to an assumption on the minimum signal, and is quite standard in the literature on learning SEM.

**Lemma 4.** *Under the assumption $\frac{a^2}{4(a^2+1)^3} \leq \mu_0 < \frac{a^2}{4}$, the Algorithm 4 outputs the global optimal solution to* (6), *i.e.*

$$\lim_{k \to \infty} W_{\mu_k} = W_{\mathsf{G}}.$$

It turns out that the assumption in Lemma 4 is not overly restrictive, as there exist pre-determined sequences of $\{\mu_k\}_{k=0}^{\infty}$ that can ensure the effectiveness of Algorithm 4 for any values of $a$ greater than a certain threshold.

## B   From Population Loss to Empirical Loss

The transformation from population loss to empirical can be thought from two components. First, with a given empirical loss, Algorithms 2 and 3 still achieve the global minimum, $W_{\mathsf{G}}$, of problem 6, but now the output from the Algorithm is an empirical estimator $\hat{a}$, rather than ground truth $a$, Theorem 1 and Corollary 1 would continue to be valid. Second, the global optimum, $W_{\mathsf{G}}$, of the empirical loss possess the same DAG structure as the underlying $W_*$. The finite-sample findings in Section 5 (specifically, Lemmas 18 and 19) of Loh and Bühlmann [31], which offer sufficient conditions on the sample size to ensure that the DAG structures of $W_{\mathsf{G}}$ and $W_*$ are identical.

## C   From Continuous to Discrete: Gradient Descent

Previously, gradient flow was employed to address the intermediate problem (7), a method that poses implementation challenges in a computational setting. In this section, we introduce Algorithm 6 that leverages gradient descent to solve (7) in each iteration. This adjustment serves practical considerations. We start with the convergence results of Gradient Descent.

**Definition 1.** *$f$ is $L$-smooth, if $f$ is differentiable and $\forall x, y \in dom(f)$ such that $\|\nabla f(x) - \nabla f(y)\|_2 \leq L\|x - y\|_2$.*

**Algorithm 5:** Gradient Descent($f, \eta, W_0, \epsilon$)

---

**Input:** function $f$, step size $\eta$, initial point $W_0$, tolerance $\epsilon$
**Output:** $W_t$
1  $t \leftarrow 0$
2  **while** $\|\nabla f(W_t)\|_2 > \epsilon$ **do**
3  $\quad$ $W_{t+1} \leftarrow W_t - \eta \nabla f(W_t)$
4  $\quad$ $t \leftarrow t + 1$
5  **end**

---

**Algorithm 6:** Homotopy algorithm using gradient descent for solving (1).

---

**Input:** Initial $W_{-1} = W(x_{-1}, y_{-1})$, $\mu_0 \in \left[ \frac{a^2}{4(a^2+1)^3} \frac{(1+\beta)^4}{(1-\beta)^2}, \frac{a^2}{4} \frac{(1-\delta)^3(1-\beta)^4}{(1+\beta)^2} \right)$,
$\quad$ $\eta_0 = \frac{1}{\mu_0(a^2+1)+3a^2}$, $\epsilon_0 = \min\{\beta a \mu_0, \mu_0^{3/2}\}$
**Output:** $\{W_{\mu_k}\}_{k=0}^{\infty}$
1  $W_{\mu_0, \epsilon_0} \leftarrow$ `Gradient Descent`$(g_{\mu_0}, \eta_0, W_{-1}, \epsilon_0)$
2  **for** $k = 1, 2, \ldots$ **do**
3  $\quad$ Let $\mu_k = (2\mu_{k-1}^2)^{2/3} \frac{(a + \epsilon_{k-1}/\mu_{k-1})^{2/3}}{(a - \epsilon_{k-1}/\mu_{k-1})^{4/3}}$
4  $\quad$ Let $\eta_k = \frac{1}{\mu_k(a^2+1)+3a^2}$
5  $\quad$ Let $\epsilon_k = \min\{\beta a \mu_k, \mu_k^{3/2}\}$
6  $\quad$ $W_{\mu_k, \epsilon_k} \leftarrow$ `Gradient Descent`$(g_{\mu_k}, \eta_k, W_{\mu_{k-1}}, \epsilon_k)$
7  **end**

---

**Theorem 3** (Nesterov et al. 33). *If function $f$ is $L$-smooth, then Gradient Descent (Algorithm 5) with step size $\eta = 1/L$, finds an $\epsilon$-first-order stationary point (i.e. $\|\nabla f(x)\|_2 \leq \epsilon$) in $2L(f(x^0) - f^*)/\epsilon^2$ iterations.*

One of the pivotal factors influencing the convergence of gradient descent is the selection of the step size. Theorem 3 select a step size $\eta = \frac{1}{L}$. Therefore, our initial step is to determine the smoothness of $g_\mu(W)$ within our region of interest, $A = \{0 \leq x \leq a, 0 \leq y \leq \frac{a}{a^2+1}\}$.

**Lemma 5.** *Consider the function $g_\mu(W)$ as defined in Equation 7 within the region $A = \{0 \leq x \leq a, 0 \leq y \leq \frac{a}{a^2+1}\}$. It follows that for all $\mu \geq 0$, the function $g_\mu(W)$ is $\mu(a^2 + 1) + 3a^2$-smooth.*

Since gradient descent is limited to identifying the $\epsilon$ stationary point of the function. Thus, we study the gradient of $g_\mu(W) = \mu f(W) + h(W)$, i.e. $\nabla g_\mu(W)$ has the following form

$$\nabla g_\mu(W) = \begin{pmatrix} \mu(x - a) + y^2 x \\ \mu(a^2 + 1)y - a\mu + yx^2 \end{pmatrix}$$

As gradient descent is limited to identifying the $\epsilon$ stationary point of the function, we, therefore, focus on $\|g_\mu(W)\|_2 \leq \epsilon$. This can be expressed in the subsequent manner:

$$\|\nabla g_\mu(W)\|_2 \leq \epsilon \Rightarrow -\epsilon \leq \mu(x - a) + y^2 x < \epsilon \quad \text{and} \quad -\epsilon \leq \mu(a^2 + 1)y - a\mu + yx^2 \leq \epsilon$$

As a result,

$$\{(x,y) \mid \|\nabla g_\mu(W)\|_2 \leq \epsilon\} \subseteq \{(x,y) \mid \frac{\mu a - \epsilon}{\mu + y^2} \leq x \leq \frac{\mu a + \epsilon}{\mu + y^2}, \frac{\mu a - \epsilon}{x^2 + \mu(a^2 + 1)} \leq y \leq \frac{\mu a + \epsilon}{x^2 + \mu(a^2 + 1)}\}$$

Here we denote such region as $A_{\mu, \epsilon}$

$$A_{\mu, \epsilon} = \{(x,y) \mid \frac{\mu a - \epsilon}{\mu + y^2} \leq x \leq \frac{\mu a + \epsilon}{\mu + y^2}, \frac{\mu a - \epsilon}{x^2 + \mu(a^2 + 1)} \leq y \leq \frac{\mu a + \epsilon}{x^2 + \mu(a^2 + 1)}\} \quad (10)$$

Figure 6 and 7 illustrate the region $A_{\mu, \epsilon}$.

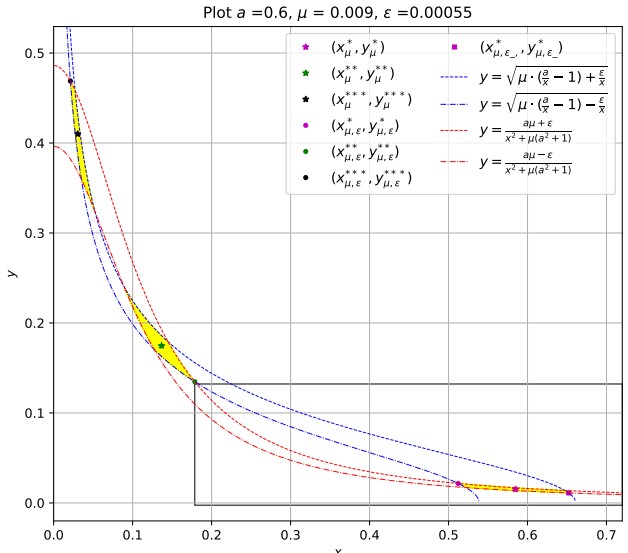

Figure 6: An example of $A_{\mu,\epsilon}$ is depicted for $a = 0.6$, $\mu = 0.009$, and $\epsilon = 0.00055$. The yellow region signifies $\epsilon$ stationary points, denoted as $A_{\mu,\epsilon}$ and defined by Equation (10). $A_{\mu,\epsilon}$ is the disjoint union of $A^1_{\mu,\epsilon}$ and $A^2_{\mu,\epsilon}$, which are defined by Equations (21) and (22), respectively.

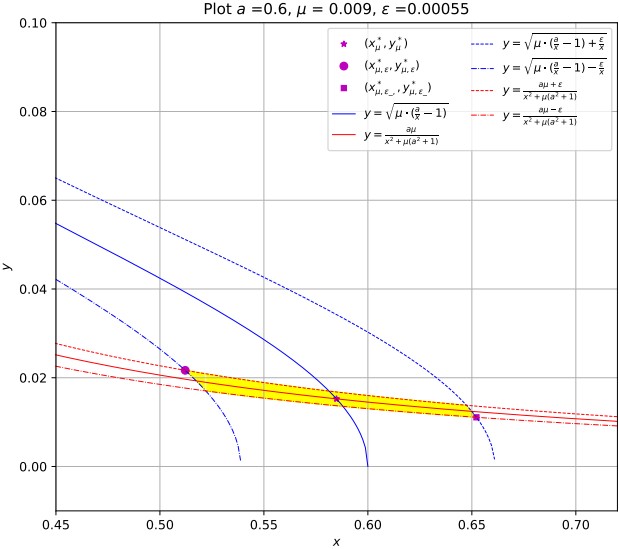

Figure 7: Here is a localized illustration of $A_{\mu,\epsilon}$ that includes the point $(x^*_\mu, y^*_\mu)$. This region, referred to as $A^1_{\mu,\epsilon}$, is defined in Equation (21).

Given that the gradient descent can only locate $\epsilon$ stationary points within the region $A_{\mu,\epsilon}$ during each iteration, the boundary of $A_{\mu,\epsilon}$ becomes a critical component of our analysis. To facilitate clear presentation, it is essential to establish some pertinent notations.

- 

$$\begin{cases} x = \dfrac{\mu a}{\mu + y^2} & \text{(11a)} \\[4mm] y = \dfrac{\mu a}{\mu(a^2 + 1) + x^2} & \text{(11b)} \end{cases}$$

If the system of equations yields only a single solution, we denote this solution as $(x_\mu^*, y_\mu^*)$. If it yields two solutions, these solutions are denoted as $(x_\mu^*, y_\mu^*), (x_\mu^{**}, y_\mu^{**})$, with $x_\mu^{**} < x_\mu^*$. In the event that there are three distinct solutions to the system of equations, these solutions are denoted as $(x_\mu^*, y_\mu^*), (x_\mu^{**}, y_\mu^{**}), (x_\mu^{***}, y_\mu^{***})$, where $x_\mu^{***} < x_\mu^{**} < x_\mu^*$.

- 

$$\begin{cases} x = \dfrac{\mu a - \epsilon}{\mu + y^2} & \text{(12a)} \\[4mm] y = \dfrac{\mu a + \epsilon}{\mu(a^2 + 1) + x^2} & \text{(12b)} \end{cases}$$

If the system of equations yields only a single solution, we denote this solution as $(x_{\mu,\epsilon}^*, y_{\mu,\epsilon}^*)$. If it yields two solutions, these solutions are denoted as $(x_{\mu,\epsilon}^*, y_{\mu,\epsilon}^*), (x_{\mu,\epsilon}^{**}, y_{\mu,\epsilon}^{**})$, with $x_{\mu,\epsilon}^{**} < x_{\mu,\epsilon}^*$. In the event that there are three distinct solutions to the system of equations, these solutions are denoted as $(x_{\mu,\epsilon}^*, y_{\mu,\epsilon}^*), (x_{\mu,\epsilon}^{**}, y_{\mu,\epsilon}^{**}), (x_{\mu,\epsilon}^{***}, y_{\mu,\epsilon}^{***})$, where $x_{\mu,\epsilon}^{***} < x_{\mu,\epsilon}^{**} < x_{\mu,\epsilon}^*$.

- 

$$\begin{cases} x = \dfrac{\mu a + \epsilon}{\mu + y^2} & \text{(13a)} \\[4mm] y = \dfrac{\mu a - \epsilon}{\mu(a^2 + 1) + x^2} & \text{(13b)} \end{cases}$$

If the system of equations yields only a single solution, we denote this solution as $(x_{\mu,\epsilon_-}^*, y_{\mu,\epsilon_-}^*)$. If it yields two solutions, these solutions are denoted as $(x_{\mu,\epsilon_-}^*, y_{\mu,\epsilon_-}^*), (x_{\mu,\epsilon_-}^{**}, y_{\mu,\epsilon_-}^{**})$, with $x_{\mu,\epsilon_-}^{**} < x_{\mu,\epsilon_-}^*$. In the event that there are three distinct solutions to the system of equations, these solutions are denoted as $(x_{\mu,\epsilon_-}^*, y_{\mu,\epsilon_-}^*), (x_{\mu,\epsilon_-}^{**}, y_{\mu,\epsilon_-}^{**}), (x_{\mu,\epsilon_-}^{***}, y_{\mu,\epsilon_-}^{***})$, where $x_{\mu,\epsilon_-}^{***} < x_{\mu,\epsilon_-}^{**} < x_{\mu,\epsilon_-}^*$.

**Remark 4.** *There always exists at least one solution to the above system of equations. When $\mu$ is sufficiently small, the above system of equations always yields three solutions, as demonstrated in Theorem 5, and Theorem 9.*

The parameter $\epsilon$ can substantially influence the behavior of the systems of equations (12a),(12b) and (13a),(13b). A crucial consideration is to ensure that $\epsilon$ remains adequately small. To facilitate this, we introduce a new parameter, $\beta$, whose specific value will be determined later. At this stage, we merely require that $\beta$ should lie within the interval $(0, 1)$. We further impose a constraint on $\epsilon$ to satisfy the following inequality:

$$\epsilon \le \beta a \mu \tag{14}$$

Following the same procedure when we deal with $\epsilon = 0$. Let us substitute (12a) into (12b), then we obtain an equation that only involves the variable $y$

$$r_\epsilon(y; \mu) = \frac{a + \epsilon/\mu}{y} - (a^2 + 1) - \frac{(\mu a - \epsilon)^2/\mu}{(y^2 + \mu)^2} \tag{15}$$

Let us substitute (12b) into (12a), then we obtain an equation that only involves the variable $x$

$$t_\epsilon(x; \mu) = \frac{a - \epsilon/\mu}{x} - 1 - \frac{(\mu a + \epsilon)^2/\mu}{(\mu(a^2 + 1) + x^2)^2} \tag{16}$$

Proceed similarly for equations (13a) and (13b).

$$r_{\epsilon_-}(y; \mu) = \frac{a - \epsilon/\mu}{y} - (a^2 + 1) - \frac{(\mu a + \epsilon)^2/\mu}{(y^2 + \mu)^2} \tag{17}$$

$$t_{\epsilon_-}(x;\mu) = \frac{a + \epsilon/\mu}{x} - 1 - \frac{(\mu a - \epsilon)^2/\mu}{(\mu(a^2+1) + x^2)^2} \tag{18}$$

Given the substantial role that the system of equations 12a and 12b play in our analysis, the existence of $\epsilon$ in these equations complicates the analysis, this can be avoided by considering the worst-case scenario, i.e., when $\epsilon = \beta a \mu$. With this particular choice of $\epsilon$, we can reformulate (15) and (16) as follows, denoting them as $r_\beta(y;\epsilon)$ and $r_\beta(x;\epsilon)$ respectively.

$$r_\beta(y;\mu) = \frac{a(1+\beta)}{y} - (a^2+1) - \frac{\mu a^2(1-\beta)^2}{(y^2+\mu)^2} \tag{19}$$

$$t_\beta(x;\mu) = \frac{a(1-\beta)}{x} - 1 - \frac{\mu a^2(1+\beta)^2}{(\mu(a^2+1)+x^2)^2} \tag{20}$$

The functions $r_\epsilon(y;\mu)$, $r_{\epsilon_-}(y;\mu)$, and $r_\beta(y;\mu)$ possess similar properties to $r(y;\mu)$ as defined in Equation (8), with more details available in Theorem 7 and 8. Additionally, the functions $t_\epsilon(x;\mu)$, $t_{\epsilon_-}(x;\mu)$, and $t_\beta(x;\mu)$ share similar characteristics with $t(x;\mu)$ as defined in Equation (9), with more details provided in Theorem 9.

As illustrated in Figure 6, the $\epsilon$-stationary point region $A_{\mu,\epsilon}$ can be partitioned into two distinct areas, of which only the lower-right one contains $(x_\mu^*, y_\mu^*)$ and it is of interest to our analysis. Moreover, $(x_{\mu,\epsilon}^*, y_{\mu,\epsilon}^*)$ and $(x_{\mu,\epsilon}^{**}, y_{\mu,\epsilon}^{**})$ are extremal point of two distinct regions. The upcoming corollary substantiates this intuition.

**Corollary 3.** If $\mu < \tau$ ($\tau$ is defined in Theorem 5(v)), assume $\epsilon$ satisfies (14), $\beta$ satisfies $\left(\frac{1+\beta}{1-\beta}\right)^2 \leq a^2 + 1$, systems of equations (12a),(12b) at least have two solutions. Moreover, $A_{\mu,\epsilon} = A_{\mu,\epsilon}^1 \cup A_{\mu,\epsilon}^2$

$$A_{\mu,\epsilon}^1 = A_{\mu,\epsilon} \cap \{(x,y) \mid x \geq x_{\mu,\epsilon}^*, y \leq y_{\mu,\epsilon}^*\} \tag{21}$$

$$A_{\mu,\epsilon}^2 = A_{\mu,\epsilon} \cap \{(x,y) \mid x \leq x_{\mu,\epsilon}^{**}, y \geq y_{\mu,\epsilon}^{**}\} \tag{22}$$

Corollary 3 suggests that $A_{\mu,\epsilon}$ can be partitioned into two distinct regions, namely $A_{\mu,\epsilon}^1$ and $A_{\mu,\epsilon}^2$. Furthermore, for every $(x,y)$ belonging to $A_{\mu,\epsilon}^1$, it follows that $x \geq x_{\mu,\epsilon}^*$ and $y \leq y_{\mu,\epsilon}^*$. Similarly, for every $(x,y)$ that lies within $A_{\mu,\epsilon}^2$, the condition $x \leq x_{\mu,\epsilon}^{**}$ and $y \geq y_{\mu,\epsilon}^{**}$ holds. The region $A_{\mu,\epsilon}^1$ represents the "correct" region that gradient descent should identify. In this context, identifying the region equates to pinpointing the extremal points of the region. As a result, our focus should be on the extremal points of $A_{\mu,\epsilon}^1$ and $A_{\mu,\epsilon}^2$, specifically at $(x_{\mu,\epsilon}^*, y_{\mu,\epsilon}^*)$ and $(x_{\mu,\epsilon}^{**}, y_{\mu,\epsilon}^{**})$. Furthermore, the key to ensuring the convergence of the gradient descent to the $A_{\mu,\epsilon}^1$ is to accurately identify the "basin of attraction" of the region $A_{\mu,\epsilon}^1$. The following lemma provides a region within which, regardless of the initialization point of the gradient descent, it converges inside $A_{\mu,\epsilon}^1$.

**Lemma 6.** Assume $\mu < \tau$ ($\tau$ is defined in Theorem 5(v)), $\left(\frac{1+\beta}{1-\beta}\right)^2 \leq a^2+1$. Define $B_{\mu,\epsilon} = \{(x,y) \mid x_{\mu,\epsilon}^{**} < x \leq a, 0 \leq y < y_{\mu,\epsilon}^{**}\}$. Run Algorithm 5 with input $f = g_\mu(x,y), \eta = \frac{1}{\mu(a^2+1)+3a^2}, W_0 = (x(0), y(0))$, where $(x(0), y(0)) \in B_{\mu,\epsilon}$, then after at most $\frac{2(\mu(a^2+1)+3a^2)(g_\mu(x(0),y(0))-g_\mu(x_\mu^*,y_\mu^*))}{\epsilon^2}$ iterations, $(x_t, y_t) \in A_{\mu,\epsilon}^1$.

Lemma 6 can be considered the gradient descent analogue of Lemma 2. It plays a pivotal role in the proof of Theorem 4. In Figure 6, the lower-right rectangle corresponds to $B_{\mu,\epsilon}$. Lemma 6 implies that the gradient descent with any initialization inside $B_{\mu_{k+1},\epsilon_{k+1}}$ will converge to $A_{\mu_{k+1},\epsilon_{k+1}}^1$ at last. Then, by utilizing the previous solution $W_{\mu_k,\epsilon_k}$ as the initial point, as long as it lies within region $B_{\mu_{k+1},\epsilon_{k+1}}$, the gradient descent can converge to $A_{\mu_{k+1},\epsilon_{k+1}}^1$ which is $\epsilon$ stationary points region that contains $W_{\mu_{k+1}}^*$, thereby achieving the goal of tracking $W_{\mu_{k+1}}^*$. Following the scheduling for $\mu_k$ prescribed in Algorithm 6 provides a sufficient condition to ensure that will happen.

We now proceed to present the theorem which guarantees the global convergence of Algorithm 6.

**Theorem 4.** If $\delta \in (0,1)$, $\beta \in (0,1)$, $\left(\frac{1+\beta}{1-\beta}\right)^2 \leq (1-\delta)(a^2+1)$, and $\mu_0$ satisfies

$$\frac{a^2}{4(a^2+1)^3} \leq \frac{a^2}{4(a^2+1)^3} \frac{(1+\beta)^4}{(1-\beta)^2} \leq \mu_0 \leq \frac{a^2}{4} \frac{(1-\delta)^3(1-\beta)^4}{(1+\beta)^2} \leq \frac{a^2}{4}$$

*Set the updating rule*

$$\epsilon_k = \min\{\beta a \mu_k, \mu_k^{3/2}\}$$

$$\mu_{k+1} = (2\mu_k^2)^{2/3} \frac{(a + \epsilon_k/\mu_k)^{2/3}}{(a - \epsilon_k/\mu_k)^{4/3}}$$

*Then $\mu_{k+1} \leq (1 - \delta)\mu_k$. Moreover, for any $\varepsilon_{\text{dist}} > 0$, running Algorithm 6 after $K(\mu_0, a, \delta, \varepsilon_{\text{dist}})$ outer iteration*

$$\|W_{\mu_k, \epsilon_k} - W_{\mathsf{G}}\|_2 \leq \varepsilon_{\text{dist}} \tag{23}$$

*where*

$$K(\mu_0, a, \delta, \varepsilon_{\text{dist}}) \geq \frac{1}{\ln(1/(1-\delta))} \max\left\{ \ln\frac{\mu_0}{\beta^2 a^2}, \ln\frac{72\mu_0}{a^2(1-(1/2)^{1/4})}, \ln(\frac{3(4-\delta)\mu_0}{\varepsilon_{\text{dist}}^2}), \frac{1}{2}\ln(\frac{46656\mu_0^2}{a^2\varepsilon_{\text{dist}}^2}), \frac{1}{3}\ln(\frac{46656\mu_0^3}{a^4\varepsilon_{\text{dist}}^2}) \right\}$$

*The total gradient descent steps are*

$$\sum_{k=0}^{K(\mu_0, a, \delta, \varepsilon_{\text{dist}})} \frac{2(\mu_k(a^2+1) + 3a^2)(g_{\mu_{k+1}}(W_{\mu_k, \epsilon_k}) - g_{\mu_{k+1}}(W_{\mu_{k+1}, \epsilon_{k+1}}))}{\epsilon_k^2}$$

$$\leq 2(\mu_0(a^2+1) + 3a^2)\left(\frac{1}{\beta^6 a^6} + \left(\max\{\frac{3(4-\delta)}{\varepsilon_{\text{dist}}^2}, \frac{216}{a\varepsilon_{\text{dist}}}, \left(\frac{216}{a\varepsilon_{\text{dist}}}\right)^{2/3}, \frac{1}{\beta^2 a^2}, \frac{72}{(1-(1/2)^{1/4})a^2}\}\right)^3\right) g_{\mu_0}(W_{\mu_0}^{\epsilon_0})$$

$$\lesssim O\left(\mu_0 a^2 + a^2 + \mu_0\right)\left(\frac{1}{\beta^6 a^6} + \frac{1}{\varepsilon_{\text{dist}}^6} + \frac{1}{a^3 \varepsilon_{\text{dist}}^3} + \frac{1}{a^2 \varepsilon_{\text{dist}}^2} + \frac{1}{a^6}\right)$$

*Proof.* Upon substituting gradient flow with gradient descent, it becomes possible to only identify an $\epsilon$-stationary point for $g_\mu(W)$. This modification necessitates specifying the stepsize $\eta$ for gradient descent, as well as an updating rule for $\mu$. The adjustment procedure can substantially influence the result of Algorithm 6. In this proof, we will impose limitations on the update scheme $\mu_k$, the stepsize $\eta_k$, and the tolerance $\epsilon_k$ to ensure their effective operation within Algorithm 6. The approach employed for this proof closely mirrors that of the proof for Theorem 1 albeit with more careful scrutiny. In this proof, we will work out all the requirements for $\mu, \epsilon, \eta$. Subsequently, we will verify that our selection in Theorem 4 conforms to these requirements.

In the proof, we occasionally use $\mu, \epsilon$ or $\mu_k, \epsilon_k$. When we employ $\mu, \epsilon$, it signifies that the given inequality or equality holds for any $\mu, \epsilon$. Conversely, when we use $\mu_k, \epsilon_k$, it indicates we are examining how to set these parameters for distinct iterations.

**Establish the Bound $y_{\mu,\epsilon}^{**} \geq \sqrt{\mu}$**   First, let us consider $r_\epsilon(\sqrt{\mu}; \mu) \leq 0$, i.e.

$$r_\epsilon(\sqrt{\mu}; \mu) = \frac{a + \epsilon/\mu}{\sqrt{\mu}} - (a^2 + 1) - \frac{\mu(a - \epsilon/\mu)^2}{4\mu^2} \leq 0$$

This is always true when $\mu > 4/a^2$, and we require

$$\epsilon \leq 2\mu^{3/2} + a\mu - 2\sqrt{2a\mu^{5/2} - \mu^3 a^2} \quad \text{when } \mu \leq \frac{4}{a^2}$$

Now we name it condition 1.

**Condition 1.**

$$\epsilon \leq 2\mu^{3/2} + a\mu - 2\sqrt{2a\mu^{5/2} - \mu^3 a^2} \quad \text{when } \mu \leq \frac{4}{a^2}$$

Under the assumption that Condition 1 is satisfied. Since $r_\epsilon(y; \mu)$ is increasing function with interval $y \in [y_{\text{lb},\epsilon}, y_{\text{ub},\epsilon}]$, and we know $y_{\text{lb},\epsilon} \leq \sqrt{\mu} \leq y_{\text{ub},\epsilon}$ and based on Theorem 7(ii), we have $y_{\text{lb},\epsilon} \leq y_{\mu,\epsilon}^{**} \leq y_{\text{ub},\epsilon}$, $r_\epsilon(\sqrt{\mu}; \mu) \leq r_\epsilon(y_{\mu,\epsilon}^{**}; \mu) = 0$. Therefore, $y_{\mu,\epsilon}^{**} \geq \sqrt{\mu}$.

**Ensuring the Correct Solution Path via Gradient Descent** Following the argument when we prove Theorem 1, we strive to ensure that the gradient descent, when initiated at $(x_{\mu_k,\epsilon_k}, y_{\mu_k,\epsilon_k})$, will converge within the "correct" $\epsilon_{k+1}$-stationary point region (namely, $\|\nabla g_{\mu_{k+1}}(W)\|_2 < \epsilon_{k+1}$) which includes $(x^*_{\mu_{k+1}}, y^*_{\mu_{k+1}})$. For this to occur, we necessitate that:

$$y_{\mu_{k+1},\epsilon_{k+1}} \overset{(1)}{>} y_{\mu_{k+1},\epsilon_{k+1}}^{**} \overset{(2)}{>} \sqrt{\mu_{k+1}} \overset{(3)}{\geq} (2\mu_k^2)^{1/3}\frac{(a+\epsilon_k/\mu_k)^{1/3}}{(a-\epsilon_k/\mu_k)^{2/3}} \overset{(4)}{>} y_{\mu_k,\epsilon_k}^* \overset{(5)}{>} y_{\mu_k,\epsilon_k} \qquad (24)$$

Here (1), (5) are due to Corollary 3; (2) comes from the boundary we established earlier; (3) is based on the constraints we have placed on $\mu_k$ and $\mu_{k+1}$, which we will present as Condition 2 subsequently; (4) is from the Theorem 7(ii) and relationship $y^*_{\mu_k,\epsilon_k} < y_{\mathrm{lb},\mu_k,\epsilon_k}$. Also, from the Lemma 9, $\max_{\mu \leq \tau} x^{**}_{\mu,\epsilon} \leq \min_{\mu>0} x^*_{\mu,\epsilon}$. Hence, by invoking Lemma 6, we can affirm that our gradient descent consistently traces the correct stationary point. Now we state condition to make it happen,

**Condition 2.**

$$(1-\delta)\mu_k \geq \mu_{k+1} \geq (2\mu_k^2)^{2/3}\frac{(a+\epsilon_k/\mu_k)^{2/3}}{(a-\epsilon_k/\mu_k)^{4/3}}$$

In this context, our requirement extends beyond merely ensuring that $\mu_k$ decreases. We further stipulate that it should decrease by a factor of $1 - \delta$. Next, we impose another important constraint

**Condition 3.**

$$\epsilon_k \leq \mu_k^{3/2}$$

**Updating Rules** Now we are ready to check our updating rules satisfy the conditions above

$$\epsilon_k = \min\{\beta a\mu_k, \mu_k^{3/2}\}$$

$$\mu_{k+1} = (2\mu_k^2)^{2/3}\frac{(a+\epsilon_k/\mu_k)^{2/3}}{(a-\epsilon_k/\mu_k)^{4/3}}$$

**Check for Conditions** First, we check the condition 2. condition 2 requires

$$(1-\delta)\mu_k \geq (2\mu_k^2)^{2/3}\frac{(a+\epsilon_k/\mu_k)^{2/3}}{(a-\epsilon_k/\mu_k)^{4/3}} \Rightarrow \mu_k\frac{(a+\epsilon_k/\mu_k)^2}{(a-\epsilon_k/\mu_k)^4} \leq \frac{(1-\delta)^3}{4}$$

Note that $\epsilon_k \leq \beta a\mu_k < a\mu_k$

$$\mu_k\frac{(a+\epsilon_k/\mu_k)^2}{(a-\epsilon_k/\mu_k)^4} \leq \mu_k\frac{(1+\beta)^2}{(1-\beta)^4}\frac{1}{a^2}$$

Therefore, once the following inequality is true, Condition 2 is satisfied.

$$\mu_k\frac{(1+\beta)^2}{(1-\beta)^4}\frac{1}{a^2} \leq \frac{(1-\delta)^3}{4} \Rightarrow \mu_k \leq \frac{a^2}{4}\frac{(1-\delta)^3(1-\beta)^4}{(1+\beta)^2}$$

Because $\mu_k \leq \mu_0 \leq \frac{a^2}{4}\frac{(1-\delta)^3(1-\beta)^4}{(1+\beta)^2}$ from the condition we impose for $\mu_0$. Consequently, Condition 2 is satisfied under our choice of $\epsilon_k$.

Now we focus on the Condition 1. Because $\epsilon_k \leq a\beta\mu_k$, if we can ensure $a\beta\mu_k \leq 2\mu_k^{3/2} + a\mu_k - 2\sqrt{2a\mu_k^{5/2} - \mu_k^3 a^2}$ holds, then we can show Condition 1 is always satisfied.

$$a\beta\mu_k \leq 2\mu_k^{3/2} + a\mu_k - 2\sqrt{2a\mu_k^{5/2} - \mu_k^3 a^2}$$

$$2\sqrt{2a\mu_k^{5/2} - \mu_k^3 a^2} \leq 2\mu_k^{3/2} + (1-\beta)a\mu_k$$

$$4(2a\mu_k^{5/2} - \mu_k^3 a^2) \leq 4\mu_k^3 + (1-\beta)^2 a^2\mu_k^2 + 4(1-\beta)a\mu_k^{5/2}$$

$$0 \leq 4(a^2+1)\mu_k^3 + (1-\beta)^2 a^2\mu_k^2 - 4(1+\beta)a\mu_k^{5/2}$$

$$0 \leq 4(a^2+1)\mu_k - 4(1+\beta)a\mu_k^{1/2} + (1-\beta)^2 a^2 \qquad \text{when} \quad 0 \leq \mu_k \leq 4/a^2$$

$$0 \leq \mu_k - \frac{(1+\beta)a}{(a^2+1)}\mu_k^{1/2} + \frac{(1-\beta)^2 a^2}{4(a^2+1)}$$

We also notice that

$$\frac{(1+\beta)^2 a^2}{(a^2+1)^2} - 4\frac{(1-\beta)^2 a^2}{4(a^2+1)} \leq 0 \Leftrightarrow \left(\frac{1+\beta}{1-\beta}\right)^2 \leq a^2 + 1$$

Because $\left(\frac{1+\beta}{1-\beta}\right)^2 \leq (1-\delta)(a^2+1)$, the inequality above always holds and this inequality implies that for any $\mu_k \geq 0$

$$0 \leq \mu_k - \frac{(1+\beta)a}{(a^2+1)}\mu_k^{1/2} + \frac{(1-\beta)^2 a^2}{4(a^2+1)}$$

Therefore, Condition 2 holds. Condition 3 also holds because of the choice of $\epsilon_k$.

**Bound the Distance**  Let $c = 72/a^2$, and assume that $\mu$ satisfies the following

$$\mu \leq \min\{\frac{1}{c}\left(1-(1/2)^{1/4}\right), \beta^2 a^2\} \tag{25}$$

Note that when $\mu$ satisfies (25), then $\mu^{3/2} \leq \beta a\mu$, so $\epsilon = \mu^{3/2}$.

$$\mu \leq \frac{1}{c}\left(1-(1/2)^{1/4}\right) = \frac{a^2}{72}\left(1-(1/2)^{1/4}\right) \leq \frac{a^2}{4}$$

$$\epsilon/\mu = \sqrt{\mu} \leq \frac{a}{2} \tag{26}$$

Then

$$\begin{aligned}
t_\epsilon((a-\epsilon/\mu)(1-c\mu);\mu) &= \frac{1}{1-c\mu} - 1 - \frac{\mu(a+\epsilon/\mu)^2}{(\mu(a^2+1)+(a-\epsilon/\mu)^2(1-c\mu)^2)^2} \\
&= \frac{c\mu}{1-c\mu} - \frac{\mu(a+\epsilon/\mu)^2}{(\mu(a^2+1)+(a-\epsilon/\mu)^2(1-c\mu)^2)^2} \\
&\geq c\mu - \mu\frac{(a+\epsilon/\mu)^2}{(a-\epsilon/\mu)^4(1-c\mu)^4} \\
&\geq c\mu - \mu\frac{(a+a/2)^2}{(a-a/2)^4(1-c\mu)^4} \\
&= \mu\left(c - \frac{36}{a^2(1-c\mu)^4}\right) \\
&= \mu\left(\frac{72}{a^2} - \frac{36}{a^2(1-c\mu)^4}\right) > 0
\end{aligned}$$

Then we know $(a-\epsilon/\mu)(1-c\mu) < x^*_{\mu,\epsilon}$. Now we can bound the distance $\|W_{\mu_k,\epsilon_k} - W_G\|$, it is important to note that

$$\begin{aligned}
\|W_{\mu_k,\epsilon_k} - W_G\| &= \sqrt{(x_{\mu_k,\epsilon_k}-a)^2 + (y_{\mu_k,\epsilon_k})^2} \\
&\leq \max\left\{\sqrt{(x^*_{\mu_k,\epsilon_k}-a)^2 + (y^*_{\mu_k,\epsilon_k})^2}, \sqrt{(x^*_{\mu_k,\epsilon_{k_-}}-a)^2 + (y^*_{\mu_k,\epsilon_k})^2}\right\}
\end{aligned}$$

We use the fact that $x^*_{\mu_k,\epsilon_k} < x_{\mu_k,\epsilon_k} < a$, $x_{\mu_k,\epsilon_k} < x^*_{\mu_k,\epsilon_{k_-}}$ and $y_{\mu_k,\epsilon_k} < y^*_{\mu_k,\epsilon_k}$. Next, we can separately establish bounds for these two terms. Due to (24), $y^*_{\mu_k,\epsilon_k} < (2\mu_k^2)^{1/3}\frac{(a+\epsilon_k/\mu_k)^{1/3}}{(a-\epsilon_k/\mu_k)^{2/3}} = \sqrt{\mu_{k+1}}$ and $(a-\epsilon_k/\mu_k)(1-c\mu_k) < x^*_{\mu_k,\epsilon_k}$

$$\sqrt{(x^*_{\mu_k,\epsilon_k}-a)^2 + (y^*_{\mu_k,\epsilon_k})^2} \leq \sqrt{\mu_{k+1} + (a-(a-\epsilon_k/\mu_k)(1-c\mu_k))^2}$$

Given that if $x^*_{\mu_k,\epsilon_{k_-}} \leq a$, then $\sqrt{(x^*_{\mu_k,\epsilon_k}-a)^2 + (y^*_{\mu_k,\epsilon_k})^2} \geq \sqrt{(x^*_{\mu_k,\epsilon_{k_-}}-a)^2 + (y^*_{\mu_k,\epsilon_k})^2}$. Therefore, if $x^*_{\mu_k,\epsilon_{k_-}} \geq a$, we can use the fact that $x^*_{\mu_k,\epsilon_{k_-}} \leq a + \frac{\epsilon_k}{\mu_k}$. In this case,

$$\sqrt{(x^*_{\mu_k,\epsilon_{k_-}}-a)^2 + (y^*_{\mu_k,\epsilon_k})^2} \leq \sqrt{\mu_{k+1}+(\epsilon_k/\mu_k)^2} = \sqrt{\mu_{k+1}+\mu_k} \leq \sqrt{(2-\delta)\mu_k}$$

As a result, we have

$$\|W_{\mu_k, \epsilon_k} - W_{\mathsf{G}}\| \le \max\{\sqrt{\mu_{k+1} + (a - (a - \epsilon_k/\mu_k)(1 - c\mu_k))^2}, \sqrt{(2 - \delta)\mu_k}\}$$

$$
\begin{aligned}
\mu_{k+1} + (a - (a - \epsilon_k/\mu_k)(1 - c\mu_k))^2 &\le (1 - \delta)\mu_k + (ac\mu_k + \sqrt{\mu_k} - c\mu_k^{3/2})^2 \\
&\le (1 - \delta)\mu_k + 3(a^2 c^2 \mu_k^2 + \mu_k + c^2 \mu_k^3) \\
&= (4 - \delta)\mu_k + 3a^2 c^2 \mu_k^2 + 3c^2 \mu_k^3
\end{aligned}
$$

$$
\begin{aligned}
\|W_{\mu_k, \epsilon_k} - W_{\mathsf{G}}\| &\le \max\{\sqrt{\mu_{k+1} + (a - (a - \epsilon_k/\mu_k)(1 - c\mu_k))^2}, \sqrt{(2 - \delta)\mu_k}\} \\
&\le \max\{\sqrt{(4 - \delta)\mu_k + 3a^2 c^2 \mu_k^2 + 3c^2 \mu_k^3}, \sqrt{(2 - \delta)\mu_k}\} \\
&= \sqrt{(4 - \delta)\mu_k + 3a^2 c^2 \mu_k^2 + 3c^2 \mu_k^3}
\end{aligned}
$$

Just let

$$(4 - \delta)\mu_k \le (4 - \delta)(1 - \delta)^k \mu_0 \le \frac{\varepsilon_{\text{dist}}^2}{3} \Rightarrow k \ge \frac{\ln(3(4 - \delta)\mu_0/\varepsilon_{\text{dist}}^2)}{\ln(1/(1 - \delta))} \tag{27}$$

$$3a^2 c^2 \mu_k^2 \le 3a^2 c^2 (1 - \delta)^{2k} \mu_0^2 \le \frac{\varepsilon_{\text{dist}}^2}{3} \Rightarrow k \ge \frac{\ln(46656\mu_0^2/(a^2 \varepsilon_{\text{dist}}^2))}{2\ln(1/(1 - \delta))} \tag{28}$$

$$3c^2 \mu_k^3 \le 3c^2 (1 - \delta)^{3k} \mu_0^3 \le \frac{\varepsilon_{\text{dist}}^2}{3} \Rightarrow k \ge \frac{\ln(46656\mu_0^3/(a^4 \varepsilon_{\text{dist}}^2))}{3\ln(1/(1 - \delta))} \tag{29}$$

We use the fact that $\mu_k \le (1 - \delta)^k \mu_0$. In order to satisfy (25).

$$\mu_k \le \mu_0 (1 - \delta)^k \le \frac{a^2}{72}(1 - (1/2)^{1/4}) \Rightarrow k \ge \frac{\ln \frac{72\mu_0}{a^2(1 - (1/2)^{1/4})}}{\ln \frac{1}{1 - \delta}} \tag{30}$$

$$\mu_k \le \mu_0 (1 - \delta)^k \le \beta^2 a^2 \Rightarrow k \ge \frac{\ln(\mu_0/(\beta^2 a^2))}{\ln \frac{1}{1 - \delta}} \tag{31}$$

Consequently, running Algorithm 6 after $K(\mu_0, a, \delta, \varepsilon_{\text{dist}})$ outer iteration

$$\|W_{\mu_k, \epsilon_k} - W_{\mathsf{G}}\|_2 \le \varepsilon_{\text{dist}}$$

where

$$K(\mu_0, a, \delta, \varepsilon_{\text{dist}}) \ge \frac{1}{\ln(1/(1 - \delta))} \max\left\{\ln \frac{\mu_0}{\beta^2 a^2}, \ln \frac{72\mu_0}{a^2(1 - (1/2)^{1/4})}, \ln(\frac{3(4 - \delta)\mu_0}{\varepsilon^2}), \frac{1}{2}\ln(\frac{46656\mu_0^2}{a^2 \varepsilon^2}), \frac{1}{3}\ln(\frac{46656\mu_0^3}{a^4 \varepsilon^2})\right\}$$

By Lemma 6, $k$ iteration of Algorithm 6 need the following step of gradient descent

$$\frac{2(\mu_k(a^2 + 1) + 3a^2)(g_{\mu_{k+1}}(W_{\mu_k, \epsilon_k}) - g_{\mu_{k+1}}(W_{\mu_{k+1}, \epsilon_{k+1}}))}{\epsilon_k^2}$$

Let $\widehat{K}(\mu_0, a, \delta, \varepsilon_{\text{dist}})$ satisfy $\mu_{\widehat{K}(\mu_0,a,\delta,\varepsilon_{\text{dist}})} \leq \beta^2 a^2 < \mu_{\widehat{K}(\mu_0,a,\delta,\varepsilon_{\text{dist}})-1}$. Hence, the total number of gradient steps required by Algorithm 6 can be expressed as follows:

$$\sum_{k=0}^{K(\mu_0,a,\delta,\varepsilon_{\text{dist}})} \frac{2(\mu_k(a^2+1)+3a^2)(g_{\mu_{k+1}}(W_{\mu_k,\epsilon_k}) - g_{\mu_{k+1}}(W_{\mu_{k+1},\epsilon_{k+1}}))}{\epsilon_k^2}$$

$$\leq 2(\mu_0(a^2+1)+3a^2)\left(\sum_{k=0}^{\widehat{K}(\mu_0,a,\delta,\varepsilon_{\text{dist}})-1} \frac{(g_{\mu_{k+1}}(W_{\mu_k,\epsilon_k}) - g_{\mu_{k+1}}(W_{\mu_{k+1},\epsilon_{k+1}}))}{\epsilon_k^2} + \sum_{k=\widehat{K}(\mu_0,a,\delta,\varepsilon_{\text{dist}})}^{K(\mu_0,a,\delta,\varepsilon_{\text{dist}})} \frac{(g_{\mu_{k+1}}(W_{\mu_k,\epsilon_k}) - g_{\mu_{k+1}}(W_{\mu_{k+1},\epsilon_{k+1}}))}{\epsilon_k^2}\right)$$

$$= 2(\mu_0(a^2+1)+3a^2)\left(\sum_{k=0}^{\widehat{K}(\mu_0,a,\delta,\varepsilon_{\text{dist}})-1} \frac{(g_{\mu_{k+1}}(W_{\mu_k,\epsilon_k}) - g_{\mu_{k+1}}(W_{\mu_{k+1},\epsilon_{k+1}}))}{\beta^2 a^2 \mu_k^2} + \sum_{k=\widehat{K}(\mu_0,a,\delta,\varepsilon_{\text{dist}})}^{K(\mu_0,a,\delta,\varepsilon_{\text{dist}})} \frac{(g_{\mu_{k+1}}(W_{\mu_k,\epsilon_k}) - g_{\mu_{k+1}}(W_{\mu_{k+1},\epsilon_{k+1}}))}{\mu_k^3}\right)$$

$$\leq 2(\mu_0(a^2+1)+3a^2)\left(\sum_{k=0}^{\widehat{K}(\mu_0,a,\delta,\varepsilon_{\text{dist}})-1} \frac{(g_{\mu_{k+1}}(W_{\mu_k,\epsilon_k}) - g_{\mu_{k+1}}(W_{\mu_{k+1},\epsilon_{k+1}}))}{\beta^6 a^6} + \sum_{k=\widehat{K}(\mu_0,a,\delta,\varepsilon_{\text{dist}})}^{K(\mu_0,a,\delta,\varepsilon_{\text{dist}})} \frac{(g_{\mu_{k+1}}(W_{\mu_k,\epsilon_k}) - g_{\mu_{k+1}}(W_{\mu_{k+1},\epsilon_{k+1}}))}{\mu_k^3}\right)$$

$$\leq 2(\mu_0(a^2+1)+3a^2)\left(\sum_{k=0}^{K(\mu_0,a,\delta,\varepsilon_{\text{dist}})} \frac{(g_{\mu_{k+1}}(W_{\mu_k,\epsilon_k}) - g_{\mu_{k+1}}(W_{\mu_{k+1},\epsilon_{k+1}}))}{\beta^6 a^6} + \sum_{k=0}^{K(\mu_0,a,\delta,\varepsilon_{\text{dist}})} \frac{(g_{\mu_{k+1}}(W_{\mu_k,\epsilon_k}) - g_{\mu_{k+1}}(W_{\mu_{k+1},\epsilon_{k+1}}))}{\mu_{K(\mu_0,a,\delta,\varepsilon_{\text{dist}})}^3}\right)$$

$$= 2(\mu_0(a^2+1)+3a^2)\left(\frac{1}{\beta^6 a^6} + \frac{1}{\mu_{K(\mu_0,a,\delta,\varepsilon_{\text{dist}})}^3}\right)\left(\sum_{k=0}^{K(\mu_0,a,\delta,\varepsilon_{\text{dist}})} \left((g_{\mu_{k+1}}(W_{\mu_k,\epsilon_k}) - g_{\mu_{k+1}}(W_{\mu_{k+1},\epsilon_{k+1}}))\right)\right)$$

$$\leq 2(\mu_0(a^2+1)+3a^2)\left(\frac{1}{\beta^6 a^6} + \frac{1}{\mu_{K(\mu_0,a,\delta,\varepsilon_{\text{dist}})}^3}\right)\left(\sum_{k=0}^{K(\mu_0,a,\delta,\varepsilon_{\text{dist}})} \left((g_{\mu_k}(W_{\mu_k}^{\epsilon_k}) - g_{\mu_{k+1}}(W_{\mu_{k+1}}^{\epsilon_{k+1}}))\right)\right)$$

$$= 2(\mu_0(a^2+1)+3a^2)\left(\frac{1}{\beta^6 a^6} + \frac{1}{\mu_{K(\mu_0,a,\delta,\varepsilon_{\text{dist}})}^3}\right)\left((g_{\mu_0}(W_{\mu_0,\epsilon_0}) - g_{\mu_{K(\mu_0,a,\delta,\varepsilon_{\text{dist}})+1}}(W_{\mu_{K(\mu_0,a,\delta,\varepsilon_{\text{dist}})+1}}^{\epsilon_{K(\mu_0,a,\delta,\varepsilon_{\text{dist}})+1}}))\right)$$

$$\leq 2(\mu_0(a^2+1)+3a^2)\left(\frac{1}{\beta^6 a^6} + \frac{1}{\mu_{K(\mu_0,a,\delta,\varepsilon_{\text{dist}})}^3}\right)g_{\mu_0}(W_{\mu_0,\epsilon_0})$$

Note from (27) and (30), the following should holds

$$\mu_{K(\mu_0,a,\delta,\varepsilon_{\text{dist}})} = \min\{\frac{\varepsilon_{\text{dist}}^2}{3(4-\delta)}, \frac{a\varepsilon_{\text{dist}}}{216}, \left(\frac{a\varepsilon_{\text{dist}}}{216}\right)^{2/3}, \beta^2 a^2, \frac{a^2}{72}(1-(1/2)^{1/4})\}$$

Therefore,

$$\sum_{k=0}^{K(\mu_0,a,\delta,\varepsilon_{\text{dist}})} \frac{2(\mu_k(a^2+1)+3a^2)(g_{\mu_{k+1}}(W_{\mu_k,\epsilon_k}) - g_{\mu_{k+1}}(W_{\mu_{k+1},\epsilon_{k+1}}))}{\epsilon_k^2}$$

$$\leq 2(\mu_0(a^2+1)+3a^2)\left(\frac{1}{\beta^6 a^6} + \left(\max\{\frac{3(4-\delta)}{\varepsilon_{\text{dist}}^2}, \frac{216}{a\varepsilon_{\text{dist}}}, \left(\frac{216}{a\varepsilon_{\text{dist}}}\right)^{2/3}, \frac{1}{\beta^2 a^2}, \frac{72}{(1-(1/2)^{1/4})a^2}\}\right)^3\right)g_{\mu_0}(W_{\mu_0}^{\epsilon_0})$$

$\square$

# D   Additional Theorems and Lemmas

**Theorem 5** (Detailed Property of $r(y;\mu)$). *For $r(y;\mu)$ in (8), then*

(i) *For $\mu > 0$, $\lim_{y\to 0^+} r(y;\mu) = \infty$, $r(\frac{a}{a^2+1},\mu) < 0$*

(ii) *For $\mu > 0$, $r(\sqrt{\mu},\mu) < 0$.*

(iii) *For $\mu > \frac{a^2}{4}$*

$$\frac{dr(y;\mu)}{dy} < 0$$

*For $0 < \mu \le \frac{a^2}{4}$*

$$\begin{cases} \dfrac{dr(y;\mu)}{dy} > 0 & y_{\text{lb}} < y < y_{\text{ub}} & \text{(32a)} \\[2mm] \dfrac{dr(y;\mu)}{dy} \le 0 & \text{Otherwise} & \text{(32b)} \end{cases}$$

*where*

$$y_{\text{lb}} = \frac{(4\mu)^{1/3}}{2}\left(a^{1/3} - \sqrt{a^{2/3} - (4\mu)^{1/3}}\right) \quad y_{\text{ub}} = \frac{(4\mu)^{1/3}}{2}\left(a^{1/3} + \sqrt{a^{2/3} - (4\mu)^{1/3}}\right)$$

*Moreover,*

$$y_{\text{lb}} \le \sqrt{\mu} \le y_{\text{ub}}$$

(iv) *For $0 < \mu < \frac{a^2}{4}$, let $p(\mu) = r(y_{\text{ub}},\mu)$, then $p'(\mu) < 0$ and there exist a unique solution to $p(\mu) = 0$, denoted as $\tau$. Additionally, $\tau < \frac{a^2}{4}$.*

(v) *There exists a $\tau > 0$ such that, $\forall \mu > \tau$, the equation $r(y;\mu) = 0$ has only one solution. At $\mu = \tau$, the equation $r(y;\mu) = 0$ has two solutions, and $\forall \mu < \tau$, the equation $r(y;\mu) = 0$ has three solutions. Moreover, $\mu < \frac{a^2}{4}$.*

(vi) *$\forall \mu < \tau$, the equation $r(y;\mu) = 0$ has three solution, i.e. $y_\mu^* < y_\mu^{**} < y_\mu^{***}$.*

$$\frac{dy_\mu^*}{d\mu} > 0 \quad \frac{dy_\mu^{**}}{d\mu} > 0 \quad \frac{dy_\mu^{***}}{d\mu} < 0 \text{ and } \lim_{\mu\to 0} y_\mu^* = 0, \lim_{\mu\to 0} y_\mu^{**} = 0, \lim_{\mu\to 0} y_\mu^{***} = \frac{a}{a^2+1}$$

*Moreover,*

$$y_\mu^* < y_{\text{lb}} < \sqrt{\mu} < y_\mu^{**} < y_{\text{ub}} < y_\mu^{***}$$

**Theorem 6** (Detailed Property of $t(x;\mu)$). *For $t(x;\mu)$ in (9), then*

(i) *For $\mu > 0$, $\lim_{x\to 0^+} t(x;\mu) = \infty$, $t(a,\mu) < 0$*

(ii) *If $\mu < \left(\frac{a(\sqrt{a^2+1}-a)}{2(a^2+1)}\right)^2$ or $\mu > \left(\frac{a(\sqrt{a^2+1}+a)}{2(a^2+1)}\right)^2$, then $t(\sqrt{\mu(a^2+1)},\mu) < 0$.*

(iii) *For $\mu > \frac{a^2}{4(a^2+1)^3}$*

$$\frac{dt(x;\mu)}{dx} < 0$$

*For $0 < \mu \le \frac{a^2}{4(a^2+1)^3}$*

$$\begin{cases} \dfrac{dt(x;\mu)}{dx} > 0 & x_{\text{lb}} < x < x_{\text{ub}} & \text{(33a)} \\[2mm] \dfrac{dt(x;\mu)}{dx} \le 0 & \text{Otherwise} & \text{(33b)} \end{cases}$$

*where*

$$x_{\text{lb}} = \frac{(4\mu a)^{1/3}(1 - \sqrt{1 - \frac{(4\mu)^{1/3}(a^2+1)}{a^{2/3}}})}{2} \quad x_{\text{ub}} = \frac{(4\mu a)^{1/3}(1 + \sqrt{1 - \frac{(4\mu)^{1/3}(a^2+1)}{a^{2/3}}})}{2}$$

*Moreover,*

$$x_{\text{lb}} \leq \sqrt{\mu(a^2 + 1)} \leq x_{\text{ub}}$$

(iv) *For $0 < \mu < \frac{a^2}{4(a^2+1)^3}$ and let $q(\mu) = t(x_{\text{lb}}, \mu)$, then $q'(\mu) > 0$ and there exist a unique solution to $q(\mu) = 0$, denoted as $\tau$ and $\tau < \frac{a^2}{4(a^2+1)^3} \leq \frac{1}{27}$.*

(v) *There exists a $\tau > 0$ such that, $\forall \mu > \tau$, the equation $t(x; \mu) = 0$ has only one solution. At $\mu = \tau$, the equation $t(x; \mu) = 0$ has two solutions, and $\forall \mu < \tau$, the equation $t(x; \mu) = 0$ has three solutions. Moreover, $\tau < \frac{a^2}{4(a^2+1)^3} \leq \frac{1}{27}$*

(vi) *$\forall \mu < \tau$, $t(x; \mu) = 0$ has three stationary points, i.e. $x_\mu^{***} < x_\mu^{**} < x_\mu^*$.*

$$\frac{dx_\mu^*}{d\mu} < 0 \quad \frac{dx_\mu^{***}}{d\mu} > 0 \text{ and } \lim_{\mu \to 0} x_\mu^* = a, \lim_{\mu \to 0} x_\mu^{**} = 0, \lim_{\mu \to 0} x_\mu^{***} = 0$$

*Besides,*

$$\max_{\mu \leq \tau} x_\mu^{**} \leq \frac{a(\sqrt{a^2 + 1} - a)}{2\sqrt{a^2 + 1}} \quad and \quad \frac{a(\sqrt{a^2 + 1} + a)}{2\sqrt{a^2 + 1}} \leq \min_{\mu > 0} x_\mu^*$$

*It also implies that $t(\frac{a(\sqrt{a^2+1}-a)}{2\sqrt{a^2+1}}; \mu) \geq 0$ and $\max_{\mu \leq \mu_0} x_\mu^{**} < \min_{\mu > 0} x_\mu^*$*

**Lemma 7.** *Algorithm 1 with input $f = g_\mu(x, y)$, $\boldsymbol{z}_0 = (x(0), y(0))$ where $(x(0), y(0)) \in C_{\mu 3}$ in (41), then $\forall t \geq 0$, $(x(t), y(t)) \in C_{\mu 3}$. Moreover, $\lim_{t \to \infty}(x(t), y(t)) = (x_\mu^*, y_\mu^*)$*

**Lemma 8.** *For any $(x, y) \in C_{\mu 3}$ in (41), and $(x, y) \neq (x_\mu^*, y_\mu^*)$*

$$g_\mu(x, y) > g_\mu(x_\mu^*, y_\mu^*)$$

**Theorem 7** (Detailed Property of $r_\epsilon(y; \mu)$)**.** *For $r_\epsilon(y; \mu)$ in (15), then*

(i) *For $\mu > 0, \epsilon > 0$, $\lim_{y \to 0^+} r_\epsilon(y; \mu) = \infty$, $y(\frac{a}{a^2+1}, \mu) < 0$*

(ii) *For $\mu > \frac{(a - \epsilon/\mu)^4}{4(a + \epsilon/\mu)^2}$, then $\frac{dr_\epsilon(y; \mu)}{dy} < 0$. For $0 < \mu \leq \frac{(a - \epsilon/\mu)^4}{4(a + \epsilon/\mu)^2}$*

$$\begin{cases} \dfrac{dr_\epsilon(y; \mu)}{dy} > 0 & y_{\text{lb},\mu,\epsilon} < y < y_{\text{ub},\mu,\epsilon} & \text{(34a)} \\[3mm] \dfrac{dr_\epsilon(y; \mu)}{dy} \leq 0 & \textit{Otherwise} & \text{(34b)} \end{cases}$$

*where*

$$y_{\text{lb},\mu,\epsilon} = \frac{(4\mu)^{1/3}}{2}\left(\left(\frac{(a - \epsilon/\mu)^2}{a - \epsilon/\mu}\right)^{1/3} - \sqrt{\left(\frac{(a - \epsilon/\mu)^2}{a - \epsilon/\mu}\right)^{2/3} - (4\mu)^{1/3}}\right)$$

$$y_{\text{ub},\mu,\epsilon} = \frac{(4\mu)^{1/3}}{2}\left(\left(\frac{(a - \epsilon/\mu)^2}{a - \epsilon/\mu}\right)^{1/3} + \sqrt{\left(\frac{(a - \epsilon/\mu)^2}{a - \epsilon/\mu}\right)^{2/3} - (4\mu)^{1/3}}\right)$$

*Also,*

$$y_{\text{lb},\mu,\epsilon} \leq (2\mu^2)^{1/3}\frac{(a + \epsilon/\mu)^{1/3}}{(a - \epsilon/\mu)^{2/3}}$$

$$y_{\text{lb},\mu,\epsilon} \leq \sqrt{\mu} \leq y_{\text{ub},\mu,\epsilon}$$

**Theorem 8** (Detailed Property of $r_\beta(y; \mu)$)**.** *For $r_\beta(y; \mu)$ in (19), then*

(i) *For $\mu > 0, \epsilon > 0, \lim_{y \to 0^+} r_\beta(y; \mu) = \infty$*

(ii) *For $\mu > \frac{a^2(1-\beta)^4}{4(1+\beta)^2}$, then $\frac{dr_\beta(y;\mu)}{dy} < 0$. For $0 < \mu \leq \frac{a^2(1-\beta)^4}{4(1+\beta)^2}$*

$$
\begin{cases}
\dfrac{dr_\beta(y;\mu)}{dy} > 0 & y_{\mathrm{lb},\mu,\beta} < y < y_{\mathrm{ub},\mu,\beta} & \text{(35a)} \\[2mm]
\dfrac{dr_\beta(y;\mu)}{dy} \leq 0 & \text{Otherwise} & \text{(35b)}
\end{cases}
$$

*where*

$$
y_{\mathrm{lb},\mu,\beta} = \frac{(4\mu)^{1/3}}{2} \left( \frac{a(1-\beta)^2}{1+\beta} \right)^{1/3} \left( 1 - \sqrt{1 - \frac{(4\mu)^{1/3}}{a^{2/3}} \left( \frac{1+\beta}{(1-\beta)^2} \right)^{2/3}} \right)
$$

$$
y_{\mathrm{ub},\mu,\beta} = \frac{(4\mu)^{1/3}}{2} \left( \frac{a(1-\beta)^2}{1+\beta} \right)^{1/3} \left( 1 + \sqrt{1 - \frac{(4\mu)^{1/3}}{a^{2/3}} \left( \frac{1+\beta}{(1-\beta)^2} \right)^{2/3}} \right)
$$

*Also,*

$$
y_{\mathrm{lb},\mu,\beta} \leq \frac{(4\mu)^{2/3}}{2a^{1/3}} \frac{(1+\beta)^{1/3}}{(1-\beta)^{2/3}}
$$

$$
y_{\mathrm{lb},\mu,\beta} \leq \sqrt{\mu} \leq y_{\mathrm{ub},\mu,\beta}
$$

**Theorem 9** (Detailed Property of $t_\beta(x;\mu)$). *For $t_\beta(x;\mu)$ in (20), then*

(i) *For $\mu > 0, \lim_{x \to 0^+} t_\beta(x;\mu) = \infty, t_\beta(a;\mu) < 0$*

(ii) *For $\mu > \frac{a^2}{4(a^2+1)^3} \frac{(\beta+1)^4}{(\beta-1)^2}$*

$$
\frac{dt_\beta(x;\mu)}{dx} < 0
$$

*For $0 < \mu \leq \frac{a^2}{4(a^2+1)^3} \frac{(\beta+1)^4}{(\beta-1)^2}$*

$$
\begin{cases}
\dfrac{dt_\beta(x;\mu)}{dx} > 0 & x_{\mathrm{lb},\mu,\beta} < x < x_{\mathrm{ub},\mu,\beta} & \text{(36a)} \\[2mm]
\dfrac{dt_\beta(x;\mu)}{dx} \leq 0 & \text{Otherwise} & \text{(36b)}
\end{cases}
$$

*where*

$$
x_{\mathrm{lb},\mu,\beta} = \frac{1}{2} \left( \frac{4a\mu(1+\beta)^2}{1-\beta} \right)^{1/3} \left( 1 - \sqrt{1 - \frac{(4\mu)^{1/3}(a^2+1)}{a^{2/3}} \left( \frac{1-\beta}{(1+\beta)^2} \right)^{2/3}} \right)
$$

$$
x_{\mathrm{ub},\mu,\beta} = \frac{1}{2} \left( \frac{4a\mu(1+\beta)^2}{1-\beta} \right)^{1/3} \left( 1 + \sqrt{1 - \frac{(4\mu)^{1/3}(a^2+1)}{a^{2/3}} \left( \frac{1-\beta}{(1+\beta)^2} \right)^{2/3}} \right)
$$

(iii) *If $0 < \beta < \frac{\sqrt{(a^2+1)}-1}{\sqrt{(a^2+1)}+1}$, then there exists a $\tau_\beta > 0$ such that, $\forall \mu > \tau_\beta$, the equation $r_\beta(x;\mu) = 0$ has only one solution. At $\mu = \tau_\beta$, the equation $r_\beta(x;\mu) = 0$ has two solutions, and $\forall \mu < \tau_\beta$, the equation $r_\beta(x;\mu) = 0$ has three solutions. Moreover, $\mu < \frac{a^2}{4(a^2+1)^3} \frac{(\beta+1)^4}{(\beta-1)^2}$.*

(iv) *If $0 < \beta < \frac{\sqrt{(a^2+1)}-1}{\sqrt{(a^2+1)}+1}$, then $\forall \mu < \tau_\beta$, $t_\beta(x;\mu) = 0$ has three stationary points, i.e. $x^{***}_{\mu,\beta} < x^{**}_{\mu,\beta} < x^{*}_{\mu,\beta}$. Besides,*

$$
\max_{\mu \leq \tau_\beta} x^{**}_{\mu,\beta} \leq \frac{a((1-\beta)\sqrt{a^2+1} - \sqrt{(1-\beta)^2(a^2+1) - (\beta+1)^2})}{2\sqrt{a^2+1}}
$$

$$
\frac{a((1-\beta)\sqrt{a^2+1} + \sqrt{(1-\beta)^2(a^2+1) - (\beta+1)^2})}{2\sqrt{a^2+1}} \leq \min_{\mu > 0} x^{*}_{\mu,\beta}
$$

*It implies that*

$$\max_{\mu \leq \tau_\beta} x^{**}_{\mu,\beta} < \min_{\mu > 0} x^*_{\mu,\beta}$$

**Lemma 9.** *Under the same setting as Corollary 3,*

$$\max_{\mu \leq \tau} x^{**}_{\mu,\epsilon} < \min_{\mu > 0} x^*_{\mu,\epsilon}$$

# E  Technical Proofs

## E.1  Proof of Theorem 3

*Proof.* For the sake of completeness, we have included the proof here. Please note that this proof can also be found in [33].

*Proof.* We use the fact that $f$ is $L$-smooth function if and only if for any $W, Y \in \text{dom}(f)$

$$f(W) \leq f(Y) + \langle \nabla f(Y), Y - W \rangle + \frac{L}{2} \|Y - W\|_2^2$$

Let $W = W^{t+1}$ and $Y = W^t$, then using the updating rule $W^{t+1} = W^t - \frac{1}{L}\nabla f(W^t)$

$$
\begin{aligned}
f(W^{t+1}) \leq & f(W^t) + \langle \nabla f(W^t), W^{t+1} - W^t \rangle + \frac{L}{2}\|W^{t+1} - W^t\|_2^2 \\
= & f(W^t) - \frac{1}{L}\|\nabla f(W^t)\|_2^2 + \frac{1}{2L}\|\nabla f(W^t)\|_2^2 \\
= & f(W^t) - \frac{1}{2L}\|\nabla f(W^t)\|_2^2
\end{aligned}
$$

Therefore,

$$\min_{0 \leq t \leq n-1} \|\nabla f(W^t)\|_2^2 \leq \frac{1}{n}\sum_{t=0}^{n-1}\|\nabla f(W^t)\|_2^2 \leq \frac{2L(f(W^0) - f(W^n))}{n} \leq \frac{2L(f(W^0) - f(W^*))}{n}$$

$$\min_{0 \leq t \leq n-1} \|\nabla f(W^t)\|_2^2 \leq \frac{2L(f(W^0) - f(W^*))}{n} \leq \epsilon^2 \Rightarrow n \geq \frac{2L(f(W^0) - f(W^*))}{\epsilon^2}$$

$\square$

$\square$

## E.2  Proof of Theorem 5

*Proof.*    (i) For any $\mu > 0$,

$$\lim_{y \to 0^+} r(y; \mu) = \lim_{y \to 0^+} \frac{a}{y} - \frac{a^2}{\mu} - (a^2 + 1) = \infty$$

$$r\left(\frac{a}{a^2 + 1}\right) = -\frac{\mu a^2}{(\frac{a}{a^2+1})^2 + \mu} < 0.$$

(ii)

$$
\begin{aligned}
r(\sqrt{\mu}, \mu) = & \frac{a}{\sqrt{\mu}} - \frac{a^2}{4\mu} - (a^2 + 1) \\
= & -\frac{a^2}{4}\left(\frac{1}{\sqrt{\mu}} - \frac{2}{a}\right)^2 - a^2 < 0
\end{aligned}
$$

(iii)

$$\frac{dr(y;\mu)}{dy} = -\frac{a}{y^2} + \frac{4a^2\mu y}{(y^2+\mu)^3}$$

$$= \frac{4a^2\mu y^3 - a(y^2+\mu)^3}{y^2(y^2+\mu)^3}$$

$$= \frac{a((4a\mu)^{2/3}y^2 + (4a\mu)^{1/3}y(y^2+\mu) + (y^2+\mu)^2)((4a\mu)^{1/3}y - y^2 - \mu)}{y^2(y^2+\mu)^3}$$

For $\mu \geq \frac{a^2}{4}$, $((4a\mu)^{1/3}y - y^2 - \mu) < 0 \Leftrightarrow \frac{dr(y;\mu)}{dy} < 0$.

For $\mu < \frac{a^2}{4}$, $y_{\mathrm{lb}} < y < y_{\mathrm{ub}}$, $((4a\mu)^{1/3}y - y^2 - \mu) > 0 \Leftrightarrow \frac{dr(y;\mu)}{dy} > 0$. For $\mu < \frac{a^2}{4}$,
$y < y_{\mathrm{lb}}$ or $y_{\mathrm{ub}} < y$, $((4a\mu)^{1/3}y - y^2 - \mu) \leq 0 \Leftrightarrow \frac{dr(y;\mu)}{dy} \leq 0$.

Note that

$$\frac{dr(y;\mu)}{d\mu} = 0 \Leftrightarrow ((4a\mu)^{1/3}y - y^2 - \mu) = 0 \Leftrightarrow (4a\mu)^{1/3} = y + \frac{\mu}{y}$$

The intersection between line $(4a\mu)^{1/3}$ and function $y + \frac{\mu}{y}$ are exactly $y_{\mathrm{lb}}$ and $y_{\mathrm{ub}}$, and $y_{\mathrm{lb}} < \sqrt{\mu} < y_{\mathrm{ub}}$.

(iv) Note that for $0 < \mu < \frac{a^2}{4}$,

$$\frac{\partial r}{\partial \mu} = -a^2 \frac{y^2 - \mu}{(\mu + y^2)^3} \quad \text{and} \quad y_{\mathrm{lb}} < \sqrt{\mu} < y_{\mathrm{ub}}$$

then $\frac{\partial r}{\partial \mu}\big|_{y=y_{\mathrm{ub}}} < 0$. Let $p(\mu) = r(y_{\mathrm{ub}}, \mu)$, because $\frac{\partial r}{\partial y}\big|_{y=y_{\mathrm{ub}}} = 0$, then

$$\frac{dp(\mu)}{d\mu} = \frac{dr(y_{\mathrm{ub}}, \mu)}{d\mu} = \frac{\partial r}{\partial y}\bigg|_{y=y_{\mathrm{ub}}} \frac{dy_{\mathrm{ub}}}{d\mu} + \frac{\partial r}{\partial \mu}\bigg|_{y=y_{\mathrm{ub}}} = \frac{\partial r}{\partial \mu}\bigg|_{y=y_{\mathrm{ub}}} < 0$$

Also note that when $\mu = \frac{a^2}{4}$, $y_{\mathrm{ub}} = \sqrt{\mu}$, $p(\mu) = r(y_{\mathrm{ub}}, \mu) = r(\sqrt{\mu}, \mu) < 0$, and also if $\mu < \frac{a^2}{4}$, then

$$y_{\mathrm{ub}} < \frac{(4\mu)^{1/3}}{2} 2a^{1/3} = (4\mu a)^{1/3}$$

Thus,

$$r((4\mu a)^{1/3}, \mu) = \frac{a}{(4\mu a)^{1/3}} - \frac{\mu a^2}{((4\mu a)^{2/3} + \mu)^2} - (a^2+1)$$

$$= \frac{a}{(4\mu a)^{1/3}} - \frac{a^2}{(\mu)^{1/3}((4a)^{2/3} + \mu^{1/3})^2} - (a^2+1)$$

$$> \frac{1}{\mu^{1/3}}\left(\frac{a}{(4a)^{1/3}} - \frac{a^2}{(4a)^{4/3}}\right) - (a^2+1)$$

Because $\frac{a}{(4a)^{1/3}} > \frac{a^2}{(4a)^{4/3}}$, it is easy to see when $\mu \to 0$, $r((4\mu a)^{1/3}, \mu) \to \infty$. We know $r(y_{\mathrm{ub}}, \mu) > r((4\mu a)^{1/3}, \mu) \to \infty$ as $\mu \to 0$ because of the monotonicity of $r(y;\mu)$ in Theorem 5(iii). Combining all of these, i.e.

$$\frac{dp(\mu)}{d\mu} < 0, \quad \lim_{\mu \to 0^+} p(\mu) = \infty, \quad p\left(\frac{a^2}{4}\right) < 0$$

There exists a $\tau < \frac{a^2}{4}$ such that $p(\tau) = 0$

(v) From Theorem 5(iv), for $\mu > \tau$, then $p(\mu) = r(y_{\mathrm{ub}}, \mu) > 0$, and for $\mu = \tau$, then $p(\mu) = r(y_{\mathrm{ub}}, \mu) = 0$. For $\mu < \tau$, then $p(\mu) = r(y_{\mathrm{ub}}, \mu) < 0$, combining Theorem 5(i),5(iii), we get the conclusions.

(vi) By Theorem 5(v), $\forall \mu < \tau$, there exists three stationary points such that $0 < y_\mu^* < y_{\mathrm{lb}} < \sqrt{\mu} < y_\mu^{**} < y_{\mathrm{ub}} < y_\mu^{***}$. Because $\left.\frac{dr(y;\mu)}{dy}\right|_{y=y_{\mathrm{lb}}} = \left.\frac{dr(y;\mu)}{dy}\right|_{y=y_{\mathrm{ub}}} = 0$, then

$$\left.\frac{dr(y;\mu)}{dy}\right|_{y=y_\mu^*} \neq 0, \quad \left.\frac{dr(y;\mu)}{dy}\right|_{y=y_\mu^{**}} \neq 0, \quad \left.\frac{dr(y;\mu)}{dy}\right|_{y=y_\mu^{***}} \neq 0$$

By implicit function theorem [14], for solution to equation $r(y;\mu) = 0$, there exists a unique continuously differentiable function such that $y = y(\mu)$ and satisfies $r(y(\mu),\mu) = 0$. Therefore,

$$\frac{\partial r}{\partial \mu} = -a^2 \frac{y^2 - \mu}{(\mu + y^2)^3}, \quad \frac{\partial r}{\partial y} = -\frac{a}{y^2} + \frac{4a^2 \mu y}{(y^2 + \mu)^3}, \quad \frac{dy(\mu)}{d\mu} = -\frac{\partial r/\partial \mu}{\partial r/\partial y}$$

Therefore by Theorem 5(iii),

$$\left.\frac{dy}{d\mu}\right|_{y=y_\mu^*} > 0 \quad \left.\frac{dy}{d\mu}\right|_{y=y_\mu^{**}} > 0 \quad \left.\frac{dy}{d\mu}\right|_{y=y_\mu^{***}} < 0$$

Because $\lim_{\mu \to 0^+} y_{\mathrm{lb}} = \lim_{\mu \to 0^+} y_{\mathrm{ub}} = 0$, then $\lim_{\mu \to 0^+} y_\mu^* = \lim_{\mu \to 0^+} y_\mu^{**} = 0$. Let us consider $r(\frac{a}{a^2+1}(1-c\mu),\mu)$ where $c = 32\frac{(a^2+1)^3}{a^2}$ and $\mu < \frac{1}{2c}$

$$r(\frac{a}{a^2+1}(1-c\mu),\mu)$$

$$= \frac{a}{\frac{a}{a^2+1}(1-c\mu)} - \frac{\mu a^2}{(\frac{a^2}{(a^2+1)^2}(1-c\mu)^2 + \mu)^2} - (a^2+1)$$

$$= (a^2+1)(\frac{c\mu}{1-c\mu}) - \frac{\mu a^2}{(\frac{a^2}{(a^2+1)^2}(1-c\mu)^2 + \mu)^2}$$

$$\geq c(a^2+1)\mu - \frac{\mu a^2}{(\frac{a^2}{(a^2+1)^2}(1-c\mu)^2)^2}$$

$$= c(a^2+1)\mu - \frac{16(a^2+1)^4}{a^2}\mu$$

$$= \frac{16(a^2+1)^4}{a^2}\mu > 0$$

By Theorem 5(iii), then $\frac{a}{a^2+1}(1-c\mu) < y_\mu^{***}$, then

$$\frac{a}{a^2+1} = \lim_{\mu \to 0^+} \frac{a}{a^2+1}(1-c\mu),\mu) \leq \lim_{\mu \to 0^+} y_\mu^{***} \leq \frac{a}{a^2+1}$$

Consequently,

$$\lim_{\mu \to 0^+} y_\mu^{***} = \frac{a}{a^2+1}$$

$\square$

### E.3 Proof of Theorem 6

*Proof.*  (i) For $\mu > 0$,

$$\lim_{x \to 0^+} t(x;\mu) = \lim_{x \to 0^+} \frac{a}{x} - \frac{a^2}{\mu(a^2+1)^2} - 1 = \infty$$

$$t(a,\mu) = -\frac{\mu a^2}{(\mu(a^2+1) + a^2)^2} < 0$$

(ii)

$$t(\sqrt{\mu(a^2+1)},\mu) = \frac{a}{\sqrt{a^2+1}}\frac{1}{\sqrt{\mu}} - \frac{a^2}{4\mu(a^2+1)^2} - 1$$

If $t(\sqrt{\mu(a^2+1)}, \mu) = 0$, then

$$\frac{1}{\sqrt{\mu}} = 2\frac{(a^2+1)^{3/2}}{a} \pm 2(a^2+1) \Rightarrow \mu = \left(\frac{a(\sqrt{a^2+1} \mp a)}{2(a^2+1)}\right)^2$$

so when $\mu < \left(\frac{a(\sqrt{a^2+1}-a)}{2(a^2+1)}\right)^2$ or $\mu > \left(\frac{a(\sqrt{a^2+1}+a)}{2(a^2+1)}\right)^2$, then $t(\sqrt{\mu(a^2+1)}, \mu) < 0$

(iii)

$$\frac{dt(x,\mu)}{dx}$$

$$= -\frac{a}{x^2} + \frac{4\mu a^2 x}{(\mu(a^2+1)+x^2)^3}$$

$$= \frac{4\mu a^2 x^3 - a(\mu(a^2+1)+x^2)^3}{x^2(\mu(a^2+1)+x^2)^3}$$

$$= \frac{a((\mu(a^2+1)+x^2)^2 + (\mu(a^2+1)+x^2)(4\mu a)^{1/3}x + (4\mu a)^{2/3}x^2)((4\mu a)^{1/3}x - \mu(a^2+1) - x^2)}{x^2(\mu(a^2+1)+x^2)^3}$$

For $\mu > \frac{a^2}{4(a^2+1)^3}$, then $(4\mu a)^{1/3}x - \mu(a^2+1) - x^2 < 0 \Leftrightarrow \frac{dt(x,\mu)}{dx} < 0$. For $\mu < \frac{a^2}{4(a^2+1)^3}$, and $x_{\mathrm{lb}} < x < x_{\mathrm{ub}}$, then $(4\mu a)^{1/3}x - \mu(a^2+1) - x^2 > 0 \Leftrightarrow \frac{dt(x,\mu)}{dx} > 0$, For $\mu < \frac{a^2}{4(a^2+1)^3}$, $x < x_{\mathrm{lb}}$ or $x > x_{\mathrm{ub}}$, $(4\mu a)^{1/3}x - \mu(a^2+1) - x^2 < 0 \Leftrightarrow \frac{dt(x,\mu)}{dx} < 0$.

We use the same argument as before to show that

$$x_{\mathrm{lb}} < \sqrt{\mu(a^2+1)} < x_{\mathrm{ub}}$$

(iv) Note that for $0 < \mu < \frac{a^2}{4(a^2+1)^3}$

$$\frac{\partial t}{\partial \mu} = -a^2 \frac{x^2 - \mu(a^2+1)}{(\mu(a^2+1)+x^2)^3} \quad \text{and} \quad x_{\mathrm{lb}} < \sqrt{\mu(a^2+1)} < x_{\mathrm{ub}}$$

then $\left.\frac{\partial t}{\partial \mu}\right|_{x=x_{\mathrm{lb}}} > 0$. Let $q(\mu) = t(x_{\mathrm{lb}}, \mu)$, because $\left.\frac{\partial t}{\partial x}\right|_{x=x_{\mathrm{lb}}} = 0$, then

$$\frac{dq(\mu)}{d\mu} = \frac{dt(x_{\mathrm{lb}}, \mu)}{d\mu} = \left.\frac{\partial t}{\partial x}\right|_{x=x_{\mathrm{lb}}}\frac{dx_{\mathrm{lb}}}{d\mu} + \left.\frac{\partial t}{\partial \mu}\right|_{x=x_{\mathrm{lb}}} = \left.\frac{\partial t}{\partial \mu}\right|_{x=x_{\mathrm{lb}}} > 0$$

Note that $\mu = \frac{a^2}{4(a^2+1)^3}$, $x_{\mathrm{ub}} = x_{\mathrm{lb}} = \frac{(4\mu a)^{1/3}}{2}$, $t(\frac{(4\mu a)^{1/3}}{2}, \frac{a^2}{4(a^2+1)^3}) = \frac{a}{(4\mu a)^{1/3}} - 1 > 0$. When $\mu < \left(\frac{a(\sqrt{a^2+1}-a)}{2(a^2+1)}\right)^2$, then $t(\sqrt{\mu(a^2+1)}, \mu) < 0$ by Theorem 6(ii). It implies that $q(\mu) < 0$ when $\mu \to 0^+$. By Theorem 6(iii), $q(\mu) = t(x_{\mathrm{lb}}, \mu) < t(\sqrt{\mu(a^2+1)}, \mu) < 0$. Combining all of the theses, i.e.

$$\frac{dq(\mu)}{d\mu} > 0, \quad \lim_{\mu \to 0^+} q(\mu) < 0, \quad q(\frac{a^2}{4(a^2+1)^3}) > 0$$

There exists a $\tau < \frac{a^2}{4(a^2+1)^3}$, $q(\tau) = 0$. Such $\tau$ is the same as in Theorem 5(iv).

(v) We follow the same proof from the proof of Theorem 5(v).

(vi) By Theorem 6(v), $\forall \mu < \mu_0$, there exists three stationary points such that $0 < x_\mu^{***} < x_{\mathrm{lb}} < x_\mu^{**} < x_{\mathrm{ub}} < x_\mu^* < a$. Because $\left.\frac{dt(x;\mu)}{dx}\right|_{x=x_{\mathrm{lb}}} = \left.\frac{dt(x;\mu)}{dx}\right|_{x=x_{\mathrm{ub}}} = 0$, then

$$\left.\frac{dt(x;\mu)}{dx}\right|_{x=x_\mu^*} \neq 0, \quad \left.\frac{dt(x;\mu)}{dx}\right|_{x=x_\mu^{**}} \neq 0, \quad \left.\frac{dt(x;\mu)}{dx}\right|_{x=x_\mu^{***}} \neq 0$$

By implicit function theorem [14], for solutions to equation $t(x; \mu) = 0$, there exists a unique continuously differentiable function such that $x = x(\mu)$ and satisfies $t(x(\mu), \mu) = 0$. Therefore,

$$\frac{dx}{d\mu} = -\frac{\partial t/\partial \mu}{\partial t/\partial x} = a^2 \frac{\frac{x^2 - \mu(a^2+1)}{(\mu(a^2+1)+x^2)^3}}{-\frac{a}{x^2} + \frac{4\mu a^2 x}{(\mu(a^2+1)+x^2)^3}}$$

Therefore, by Theorem 6(iii)

$$\left.\frac{dx}{d\mu}\right|_{x=x_\mu^*} < 0 \quad \left.\frac{dx}{d\mu}\right|_{x=x_\mu^{***}} > 0$$

Because $0 < x_\mu^{***} < x_{\text{lb}} < x_\mu^{**} < x_{\text{ub}}$ and $\lim_{\mu\to 0^+} x_{\text{lb}} = \lim_{\mu\to 0^+} x_{\text{ub}} = 0$.

$$\lim_{\mu\to 0} x_\mu^{**} = \lim_{\mu\to 0} x_\mu^{***} = 0$$

Let us consider $t(a(1 - c\mu), \mu)$ where $c = \frac{32}{a^2}$ and $\mu < \frac{1}{2c}$

$$t(a(1 - c\mu); \mu)$$

$$= \frac{a}{a(1 - c\mu)} - \frac{\mu a^2}{(\mu(a^2 + 1) + a^2(1 - c\mu)^2)^2} - 1$$

$$= \frac{c\mu}{1 - c\mu} - \frac{\mu a^2}{(\mu(a^2 + 1) + a^2(1 - c\mu)^2)^2}$$

$$\geq c\mu - \frac{\mu a^2}{(a^2(1 - c\mu)^2)^2}$$

$$\geq c\mu - \frac{16}{a^2}\mu > 0$$

By Theorem 6(iii). It implies

$$a(1 - c\mu) \leq x_\mu^*$$

taking $\mu \to 0^+$ on both side,

$$a = \lim_{\mu\to 0^+} a(1 - c\mu) \leq \lim_{\mu\to 0^+} x_\mu^* \leq a$$

Hence, $\lim_{\mu\to 0} x_\mu^* = a$.

When $\mu = \tau$, because $t(x_{\text{lb}}; \mu) = 0$ and $x_{\text{ub}} > \sqrt{\mu(a^2 + 1)} > x_{\text{lb}}$, $t(x; \mu)$ is increasing function between $[x_{\text{lb}}, x_{\text{ub}}]$ then $t(\sqrt{\mu(a^2 + 1)}; \mu) > t(x_{\text{lb}}; \mu) = 0$. Moreover, $t(\sqrt{\mu(a^2 + 1)}, \mu)$, $x_{\text{lb}}$ and $x_\mu^{**}$ are continuous function w.r.t $\mu$, $\exists \delta > 0$ which is really small, such that $\mu = \tau - \delta$ and $t(\sqrt{\mu(a^2 + 1)}, \mu) > 0$, $t(x_{\text{lb}}, \mu) < 0$ (by Theorem 6(iv)) and $x_\mu^{**} > x_{\text{lb}}$, hence $\left.\frac{dx}{d\mu}\right|_{x=x_\mu^{**}} < 0$. It implies when $\mu$ decreases, then $x_\mu^{**}$ increases. This relation holds until $x_\mu^{**} = \sqrt{\mu(a^2 + 1)}$

$$t(x_\mu^{**}, \mu) = t(\sqrt{\mu(a^2 + 1)}, \mu) = 0$$

$$\Rightarrow \mu = \left(\frac{a(\sqrt{a^2 + 1} - a)}{2(a^2 + 1)}\right)^2$$

and $\sqrt{\mu(a^2 + 1)} = \frac{a(\sqrt{a^2+1}-a)}{2\sqrt{a^2+1}}$. Note that when $\mu < \left(\frac{a(\sqrt{a^2+1}-a)}{2(a^2+1)}\right)^2$, $t(\sqrt{\mu(a^2 + 1)}, \mu) < 0$, it implies that $x_\mu^{**} > \sqrt{\mu(a^2 + 1)}$ and $\left.\frac{dx}{d\mu}\right|_{x=x_\mu^{**}} > 0$, thus decreasing $\mu$ leads to decreasing $x_\mu^{**}$. We can conclude

$$\max_{\mu\leq\tau} x_\mu^{**} \leq \frac{a(\sqrt{a^2 + 1} - a)}{2\sqrt{a^2 + 1}}$$

Note that $\forall \mu$ s.t. $\left(\frac{a(\sqrt{a^2+1}-a)}{2(a^2+1)}\right)^2 < \mu < \tau$, $x_\mu^{**} < \left(\frac{a(\sqrt{a^2+1}-a)}{2(a^2+1)}\right)^2$, so $t\left(\left(\frac{a(\sqrt{a^2+1}-a)}{2(a^2+1)}\right)^2, \mu\right) \geq 0$.

Note that when $\mu > \frac{a^2}{a^2+1}$, i.e. $(x_\mu^*)^2 \geq \mu(a^2+1)$ then

$$\left.\frac{dx}{d\mu}\right|_{x=x_\mu^*} > 0$$

It implies that when $\mu$ decreases, $x_\mu^*$ also decreases. It holds true until $x_\mu^* = \sqrt{\mu(a^2+1)}$. The same analysis can be applied to $x_\mu^*$ like above, we can conclude that

$$\min_\tau x_\mu^* = \frac{a(\sqrt{a^2+1}+a)}{2\sqrt{a^2+1}}$$

Hence

$$\max_{\mu \leq \tau} x_\mu^{**} \leq \frac{a(\sqrt{a^2+1}-a)}{2\sqrt{a^2+1}} < \frac{a(\sqrt{a^2+1}+a)}{2\sqrt{a^2+1}} \leq \min_{\mu>0} x_\mu^*$$

$\square$

## E.4 Proof of Theorem 7,8 and 9

*Proof.* The proof is similar to the proof of Theorem 5 and Theorem 6. $\square$

## E.5 Proof of Lemma 1

*Proof.*

$$\nabla^2 g_\mu(x,y) = \begin{pmatrix} \mu + y^2 & 2xy \\ 2xy & \mu(a^2+1) + x^2 \end{pmatrix}$$

Let $\lambda_1(\nabla^2 g_\mu(x,y))$, $\lambda_2(\nabla^2 g_\mu(x,y))$ be the eigenvalue of matrix $\nabla^2 g_\mu(x,y)$, then

$$\lambda_1(\nabla^2 g_\mu(x,y)) + \lambda_2(\nabla^2 g_\mu(x,y))$$
$$= \mathrm{Tr}(\nabla^2 g_\mu(x,y)) = \mu + y^2 + \mu(a^2+1) + x^2 > 0$$

Now we calculate the product of eigenvalue

$$\lambda_1(\nabla^2 g_\mu(x,y)) \cdot \lambda_2(\nabla^2 g_\mu(W))$$
$$= \det(\nabla^2 g_\mu(W))$$
$$= (\mu + y^2)(\mu(a^2+1) + x^2) - 4x^2 y^2$$
$$= \frac{\mu a}{x}\frac{\mu a}{y} - 4x^2 y^2 > 0$$
$$\Leftrightarrow (\frac{a\mu}{2})^{2/3} > xy$$
$$\Leftrightarrow (\frac{a\mu}{2})^{2/3} > \frac{a\mu}{y^2+\mu}y$$
$$\Leftrightarrow y + \frac{\mu}{y} > (4a\mu)^{1/3}$$

Note that for $(x_\mu^*, y_\mu^*), (x_\mu^{***}, y_\mu^{***})$, they satisfy (11a) and (11b), this fact is used in third equality and second "$\Leftrightarrow$". By (32b), we know $\lambda_1(\nabla^2 g_\mu(x,y)) \cdot \lambda_2(\nabla^2 g_\mu(x,y)) > 0$ for $(x_\mu^*, y_\mu^*), (x_\mu^{***}, y_\mu^{***})$, and $\lambda_1(\nabla^2 g_\mu(x,y)) \cdot \lambda_2(\nabla^2 g_\mu(x,y)) < 0$ for $(x_\mu^{**}, y_\mu^{**})$, then

$$\lambda_1(\nabla^2 g_\mu(x,y)) > 0, \lambda_2(\nabla^2 g_\mu(x,y)) > 0 \qquad \text{for } (x_\mu^*, y_\mu^*), (x_\mu^{***}, y_\mu^{***})$$
$$\lambda_1(\nabla^2 g_\mu(x,y)) < 0 \text{ or } \lambda_2(\nabla^2 g_\mu(x,y)) < 0 \qquad \text{for } (x_\mu^{**}, y_\mu^{**})$$

and

$$\nabla g_\mu(x,y) = 0$$

Then $(x_\mu^*, y_\mu^*), (x_\mu^{***}, y_\mu^{***})$ are locally minima, $(x_\mu^{**}, y_\mu^{**})$ is saddle point for $g_\mu(W)$. $\square$

## E.6 Proof of Lemma 2

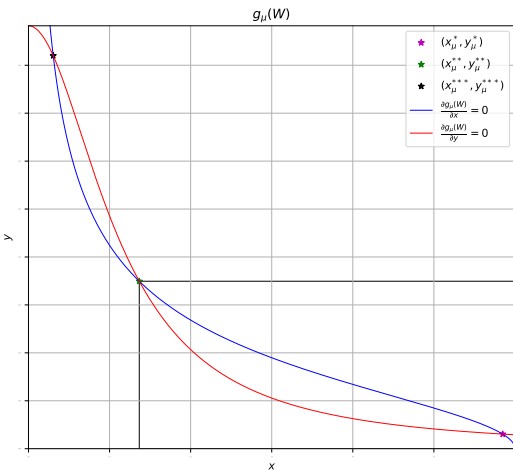

Figure 8: Stationary points when $\mu < \tau$

*Proof.* Let us define the functions as below

$$
\begin{cases}
y_{\mu 1}(x) = \sqrt{\mu(\dfrac{a-x}{x})} & 0 < x \le a \\[3mm]
y_{\mu 2}(x) = \dfrac{\mu a}{\mu(a^2+1)+x^2} & 0 < x \le a
\end{cases}
$$
(37a)
(37b)

$$
\begin{cases}
x_{\mu 1}(y) = \dfrac{\mu a}{y^2+\mu} & 0 < y < \frac{a}{a^2+1} \\[3mm]
x_{\mu 2}(y) = \sqrt{\mu(\dfrac{a}{y}-(a^2+1))} & 0 < y < \frac{a}{a^2+1}
\end{cases}
$$
(38a)
(38b)

with simple calculations,

$$ y_{\mu 1} \ge y_{\mu 2} \Leftrightarrow t(x;\mu) \ge 0 \Leftrightarrow x \in (0, x_\mu^{***}] \cup [x_\mu^{**}, x_\mu^*] $$

and

$$ x_{\mu 1} \ge x_{\mu 2} \Leftrightarrow r(y;\mu) \le 0 \Leftrightarrow y \in [y_\mu^*, y_\mu^{**}] \cup [y_\mu^{***}, \frac{a}{a^2+1}) $$

Here we divide $B_\mu$ into three parts, $C_{\mu 1}, C_{\mu 2}, C_{\mu 3}$

$$ C_{\mu 1} = \{(x,y)|x_\mu^{**} < x \le x_\mu^*, y_{\mu 1} < y < y_\mu^{**}\} \cup \{(x,y)|x_\mu^* < x \le a, y_{\mu 2} < y < y_\mu^{**}\} $$
(39)
$$ C_{\mu 2} = \{(x,y)|x_\mu^{**} < x \le x_\mu^*, 0 \le y < y_{\mu 2}\} \cup \{(x,y)|x_\mu^* < x \le a, 0 \le y < y_{\mu 1}\} $$
(40)
$$ C_{\mu 3} = \{(x,y)|x_\mu^{**} < x \le x_\mu^*, y_{\mu 2} \le y \le y_{\mu 1}\} \cup \{(x,y)|x_\mu^* < x \le a, y_{\mu 1} \le y \le y_{\mu 2}\} $$
(41)

Also note that

$$ \forall (x,y) \in C_{\mu 1} \Rightarrow \frac{\partial g_\mu(x,y)}{\partial x} > 0, \frac{\partial g_\mu(x,y)}{\partial y} > 0 $$

$$ \forall (x,y) \in C_{\mu 2} \Rightarrow \frac{\partial g_\mu(x,y)}{\partial x} < 0, \frac{\partial g_\mu(x,y)}{\partial y} < 0 $$

The gradient flow follows

$$ \begin{pmatrix} x'(t) \\ y'(t) \end{pmatrix} = - \begin{pmatrix} \frac{\partial g_\mu(x(t),y(t))}{\partial x} \\ \frac{\partial g_\mu(x(t),y(t))}{\partial y} \end{pmatrix} = -\nabla g_\mu(x(t), y(t)) $$

then

$$\forall (x,y) \in C_{\mu 1} \Rightarrow \begin{pmatrix} x'(t) \\ y'(t) \end{pmatrix} < 0, \quad \|\nabla g_\mu\| > 0 \tag{42}$$

$$\forall (x,y) \in C_{\mu 2} \Rightarrow \begin{pmatrix} x'(t) \\ y'(t) \end{pmatrix} > 0, \quad \|\nabla g_\mu\| > 0 \tag{43}$$

Note that $\|\nabla g_\mu\|$ is not diminishing and bounded away from 0. Let us consider the $(x(0), y(0)) \in C_{\mu 1}$, since $\nabla g_\mu(x,y) \neq 0$, $-\nabla g_\mu(x,y) < 0$ in (42) and boundness of $C_{\mu 1}$, it implies there exists a finite $t_0 > 0$ such that

$$(x(t_0), y(t_0)) \in \partial C_{\mu 1}, (x(t), y(t)) \in C_{\mu 1} \text{ for } 0 \leq t < t_0$$

where $\partial C_{\mu 1}$ is defined as

$$\partial C_{\mu 1} = \{(x,y)|x_\mu^{**} < x \leq x_\mu^*, y = y_{\mu 1}\} \cup \{(x,y)|x_\mu^* < x \leq a, y = y_{\mu 2}\} \subseteq C_{\mu 3}$$

For the same reason, if $(x(0), y(0)) \in C_{\mu 2}$, there exists a finite time $t_1 > 0$,

$$(x(t_0), y(t_0)) \in \partial C_{\mu 2}, (x(t), y(t)) \in C_{\mu 2} \text{ for } 0 \leq t < t_1$$

where $\partial C_{\mu 2}$ is defined as

$$\partial C_{\mu 2} = \{(x,y)|x_\mu^{**} < x \leq x_\mu^*, y = y_{\mu 2}\} \cup \{(x,y)|x_\mu^* < x \leq a, y = y_{\mu 1}\} \subseteq C_{\mu 3}$$

then by lemma 7, $\lim_{t \to \infty}(x(t), y(t)) = (x_\mu^*, y_\mu^*)$. □

### E.7 Proof of Lemma 3

*Proof.* This is just a result of the Theorem 5. □

### E.8 Proof of Lemma 5

*Proof.* Note that

$$\nabla^2 g_\mu(W) = \begin{pmatrix} \mu + y^2 & 2xy \\ 2xy & \mu(a^2+1) + x^2 \end{pmatrix} = \begin{pmatrix} \mu & 0 \\ 0 & \mu(a^2+1) \end{pmatrix} + \begin{pmatrix} y^2 & 2xy \\ 2xy & x^2 \end{pmatrix}$$

Let $\|\cdot\|_{op}$ is the spectral norm, and it satisfies triangle inequality

$$\left\|\nabla^2 g_\mu(W)\right\|_{op} \leq \left\|\begin{pmatrix} \mu & 0 \\ 0 & \mu(a^2+1) \end{pmatrix}\right\|_{op} + \left\|\begin{pmatrix} y^2 & 2xy \\ 2xy & x^2 \end{pmatrix}\right\|_{op}$$

$$= \mu(a^2+1) + \left\|\begin{pmatrix} y^2 & 2xy \\ 2xy & x^2 \end{pmatrix}\right\|_{op}$$

The spectral norm of the second term in area A is bounded by

$$\max_{(x,y) \in A} \frac{(x^2+y^2) + \sqrt{(x^2+y^2)^2 + 12x^2y^2}}{2} \leq \frac{2a^2 + \sqrt{4a^4 + 12a^4}}{2} = 3a^2$$

We use $x^2 \leq a^2, y^2 \leq a^2$ in the inequality. Therefore,

$$\left\|\nabla^2 g_\mu(W)\right\|_{op} \leq 3a^2 + \mu(a^2+1)$$

Also, according to [5, 33], for any $f$, if $\nabla^2 f$ exists, then $f$ is $L$ smooth if and only if $|\nabla^2 f|_{op} \leq L$. With this, we conclude the proof. □

### E.9 Proof of Lemma 7

*Proof.* First we prove $\forall t \geq 0, (x(t), y(t)) \in C_{\mu 3}$, because if $(x(t), y(t)) \notin C_{\mu 3}$, then there exists a finite $t$ such that

$$(x(t), y(t)) \in \partial C_{\mu 3}$$

where $\partial C_{\mu 3}$ is the boundary of $C_{\mu 3}$, defined as

$$\partial C_{\mu 3} = \{(x,y)|y = y_{\mu 1}(x) \text{ or } y = y_{\mu 2}(x), x_\mu^{**} < x \leq a\}$$

W.L.O.G, let us assume $(x(0), y(0)) \in \partial C_{\mu 3}$ and $(x(0), y(0)) \neq (x_\mu^*, y_\mu^*)$. Here are four different cases,

$$\nabla g_\mu(x(t), y(t)) = \begin{cases} \begin{pmatrix} = 0 \\ > 0 \end{pmatrix} & \text{if } y(0) = y_{\mu 1}(x(0)), x_\mu^{**} < x(0) < x_\mu^* \\ \begin{pmatrix} = 0 \\ < 0 \end{pmatrix} & \text{if } y(0) = y_{\mu 1}(x(0)), x_\mu^* < x(0) \leq a \\ \begin{pmatrix} < 0 \\ = 0 \end{pmatrix} & \text{if } y(0) = y_{\mu 2}(x(0)), x_\mu^{**} < x(0) < x_\mu^* \\ \begin{pmatrix} > 0 \\ = 0 \end{pmatrix} & \text{if } y(0) = y_{\mu 2}(x(0)), x_\mu^* < x(0) \leq a \end{cases}$$

This indicates that $-\nabla g_\mu(x(t), y(t))$ are pointing to the interior of $C_{\mu 3}$, then $(x(t), y(t))$ can not escape $C_{\mu 3}$. Here we can focus our attention in $C_{\mu 3}$, because $\forall t \geq 0, (x(t), y(t)) \in C_{\mu 3}$. For Algorithm 1,

$$\frac{df(\mathbf{z}_t)}{dt} = \nabla f(\mathbf{z}_t)\dot{\mathbf{z}}_t = -\|\nabla f(\mathbf{z}_t)\|_2^2$$

In our setting, $\forall (x, y) \in C_{\mu 3}$

$$\begin{cases} \nabla g_\mu(x, y) \neq 0 & (x, y) \neq (x_\mu^*, y_\mu^*) \\ \nabla g_\mu(x, y) = 0 & (x, y) = (x_\mu^*, y_\mu^*) \end{cases}$$

so

$$\frac{dg_\mu(x(t), y(t))}{dt} = \begin{cases} -\|\nabla g_\mu\|_2^2 < 0 & (x, y) \neq (x_\mu^*, y_\mu^*) \\ -\|\nabla g_\mu\|_2^2 = 0 & (x, y) = (x_\mu^*, y_\mu^*) \end{cases}$$

Plus, $(x_\mu^*, y_\mu^*)$ is the unique stationary point of $g_\mu(W)$ in $C_{\mu 3}$. By lemma 8

$$g_\mu(x, y) > g_\mu(x_\mu^*, y_\mu^*) \quad (x, y) \neq (x_\mu^*, y_\mu^*)$$

By Lyapunov asymptotic stability theorem [28], and applying it to gradient flow for $g_\mu(x, y)$ in $C_{\mu 3}$, we can conclude $\lim_{t \to \infty}(x(t), y(t)) = (x_\mu^*, y_\mu^*)$. □

### E.10 Proof of Lemma 8

*Proof.* For any $(x, y) \in C_{\mu 3}$ in 41, and $(x, y) \neq (x_\mu^*, y_\mu^*)$, in Algorithm 7. W.L.O.G, we can assume $x \in (x_\mu^{**}, x_\mu^*)$, the analysis details can also be applied to $x \in (x_\mu^*, a)$. It is obvious that $\tilde{x}_j < \tilde{x}_{j+1}$ and $\tilde{y}_{j+1} < \tilde{y}_j$. Also, $\lim_{j \to \infty}(\tilde{x}_j, \tilde{y}_j) = (x_\mu^*, y_\mu^*)$. Otherwise either $\tilde{x}_j \neq x_\mu^*$ or $\tilde{y}_j \neq y_\mu^*$ hold, Algorithm 7 continues until $\lim_{j \to \infty}(\tilde{x}_j, \tilde{y}_j) = \lim_{j \to \infty}(y_{\mu 2}(\tilde{y}_j), x_{\mu 1}(\tilde{x}_j))$, i.e. $(\tilde{x}_j, \tilde{y}_j)$ converges to $(x_\mu^*, y_\mu^*)$.

Moreover, note that for any $j = 0, 1, \ldots$

$$g_\mu(\tilde{x}_{j-1}, \tilde{y}_{j-1}) > g_\mu(\tilde{x}_{j-1}, \tilde{y}_j) > g_\mu(\tilde{x}_j, \tilde{y}_j)$$

Because

$$g_\mu(\tilde{x}_{j-1}, \tilde{y}_{j-1}) - g_\mu(\tilde{x}_{j-1}, \tilde{y}_j) = \frac{\partial g_\mu(\tilde{x}_{j-1}, \tilde{y})}{\partial y}(\tilde{y}_{j-1} - \tilde{y}_j) \quad \text{where } \tilde{y} \in (\tilde{y}_j, \tilde{y}_{j-1})$$

Note that

$$\frac{\partial g_\mu(\tilde{x}_{j-1}, \tilde{y})}{\partial y} > 0 \Rightarrow g_\mu(\tilde{x}_{j-1}, \tilde{y}_{j-1}) > g_\mu(\tilde{x}_{j-1}, \tilde{y}_j)$$

By the same reason,

$$g_\mu(\tilde{x}_{j-1}, \tilde{y}_j) > g_\mu(\tilde{x}_j, \tilde{y}_j)$$

By Lemma 1, $(x_\mu^*, y_\mu^*)$ is local minima, and there exists a $r_\mu > 0$ and any $\{(x, y) \mid \|(x, y) - (x_\mu^*, y_\mu^*)\|_2 \leq r_\mu\}, g_\mu(x, y) > g_\mu(x_\mu^*, y_\mu^*)$ Since $\lim_{j \to \infty}(\tilde{x}_j, \tilde{y}_j) = (x_\mu^*, y_\mu^*)$, there exists a $J > 0$ such that $\forall j > J, \|(\tilde{x}_j, \tilde{y}_j) - (x_\mu^*, y_\mu^*)\|_2 \leq r_\mu$, combining them all

$$g_\mu(x, y) > g_\mu(\tilde{x}_j, \tilde{y}_j) > g_\mu(x_\mu^*, y_\mu^*)$$

□

**Algorithm 7:** Path goes to $(x_\mu^*, y_\mu^*)$

---

**Input:** $(x, y) \in C_{\mu 3}, x_{\mu 1}(y), y_{\mu 2}(x)$ as (38a),(37b)
**Output:** $\{(\tilde{x}_j, \tilde{y}_j)\}_{j=0}^\infty$
1 $(\tilde{x}_0, \tilde{y}_0) \leftarrow (x, y)$
2 **for** $j = 1, 2, \ldots$ **do**
3     $\tilde{y}_j \leftarrow y_{\mu 2}(\tilde{x}_{j-1})$
4     $\tilde{x}_j \leftarrow x_{\mu 1}(\tilde{y}_{j-1})$
5 **end**

---

### E.11 Proof of Lemma 4

*Proof.* From the proof of Theorem 1, any any scheduling for $\mu_k$ satisfies following will do the job

$$(2/a)^{2/3} \mu_{k-1}^{4/3} \leq \mu_k < \mu_{k-1}$$

Note that in Algorithm 4, we have $\hat{a} = \sqrt{4(\mu_0 + \varepsilon)} < a$, then it is obvious

$$(2/a)^{2/3} \mu_{k-1}^{4/3} < (2/\hat{a})^{2/3} \mu_{k-1}^{4/3}$$

The same analysis for Theorem 1 can be applied here. $\qquad\square$

### E.12 Proof of Lemma 6

*Proof.* By the Theorem 3 and Lemma 5 and the fact that $A_{\mu, \epsilon}^1$ is $\mu$-stationary point region, we use the same argument as proof of Lemma 7 to demonstrate the gradient descent will never go to $A_{\mu, \epsilon}^2$. $\quad\square$

### E.13 Proof of Lemma 9

*Proof.* By Theorem 9(iv)

$$\max_{\mu \leq \tau_\beta} x_{\mu, \beta}^{**} \leq \min_{\mu > 0} x_{\mu, \beta}^*$$

We also know from the proof of Corollary 3, $x_{\mu, \epsilon}^{**} < x_{\mu, \beta}^{**}$ and $x_{\mu, \beta}^* < x_{\mu, \epsilon}^*$. Consequently,

$$\max_{\mu \leq \tau_\beta} x_{\mu, \epsilon}^{**} \leq \min_{\mu > 0} x_{\mu, \epsilon}^*$$

Because $\tau_\beta > \tau$, so

$$\max_{\mu \leq \tau} x_{\mu, \epsilon}^{**} \leq \max_{\mu \leq \tau_\beta} x_{\mu, \epsilon}^{**} \leq \min_{\mu > 0} x_{\mu, \epsilon}^*$$

$\qquad\square$

### E.14 Proof of Corollary 1

*Proof.* Note that

$$\frac{a^2}{4(a^2 + 1)^3} \leq \frac{1}{27} \quad a > 0$$

when $a > \sqrt{\frac{5}{27}}$, then $\frac{a^2}{4} > \mu_0 = \frac{1}{27} \geq \frac{a^2}{4(a^2+1)^3}$, it satisfies condition in Lemma 4, we obtain the same result. $\qquad\square$

### E.15 Proof of Corollary 2

*Proof.* Use Theorem 5(vi) and Theorem 6(vi). $\qquad\square$

### E.16 Proof of Corollary 3

*Proof.* It is easy to know that

$$r_\beta(y; \mu) > r_\epsilon(y; \mu) > r(y; \mu)$$

and

$$t_\beta(x; \mu) < t_\epsilon(x; \mu) < t(x; \mu)$$

and when $\mu < \tau$, there are three solutions to $r(y; \mu) = 0$ by Theorem 5. Also, we know from Theorem 7, 8

$$\lim_{y \to 0^+} r_\epsilon(y; \mu) = \infty \qquad \lim_{y \to 0^+} r_\beta(y; \mu) = \infty$$

Note that when $\left(\frac{1+\beta}{1-\beta}\right)^2 \le a^2 + 1$

$$r_\beta(\sqrt{\mu}; \mu) = \frac{a(1+\beta)}{\sqrt{\mu}} - (a^2 + 1) - \frac{a^2(1-\beta)^2}{4\mu} \le 0 \quad \forall \mu > 0$$

Therefore,

$$0 \ge r_\beta(\sqrt{\mu}; \mu) > r_\epsilon(\sqrt{\mu}; \mu) > r(\sqrt{\mu}; \mu)$$

Also, we know that for $y_{\mathrm{ub}}$ defined in Theorem 5(iii), we know $r(y_{\mathrm{ub}}; \mu) > 0$ from Theorem 5(iv). Therefore,

$$r_\beta(y_{\mathrm{ub}}; \mu) > r_\epsilon(y_{\mathrm{ub}}; \mu) > r(y_{\mathrm{ub}}; \mu) > 0$$

Besides, $\sqrt{\mu} < y_{\mathrm{ub}}$. By monotonicity of $r_\beta(y; \mu)$ and $r_\epsilon(y; \mu)$ from the Theorem 7(ii) and Theorem 8(ii), it implies that there are at least two solutions to $r_\beta(y; \mu)$ and $r_\epsilon(y; \mu)$. From the geometry of $r_\beta(y; \mu), r_\epsilon(y; \mu), r(y; \mu)$ and $t_\beta(x; \mu), t_\epsilon(x; \mu), t(x; \mu)$, it is trivial to know that $x^*_{\mu, \epsilon} \le x^*_\mu$, $y^*_{\mu, \epsilon} \ge y^*_\mu$, $x^{**}_{\mu, \epsilon} \ge x^{**}_\mu$, $y^*_{\mu, \epsilon} \le y^{**}_\mu$.

Finally, for every point $(x, y) \in A^1_{\mu, \epsilon}$, there exists a pair $\epsilon_1, \epsilon_2$, each satisfying $|\epsilon_1| \le \epsilon$ and $|\epsilon_2| \le \epsilon$, such that $(x, y)$ is the solution to

$$x = \frac{\mu a + \epsilon_1}{\mu + y^2} \qquad y = \frac{\mu a + \epsilon_2}{x^2 + \mu(a^2 + 1)}$$

We can repeat the same analysis above to show that $x^*_{\mu, \epsilon} \le x, y^*_{\mu, \epsilon} \ge y$. Applying the same logic to $\forall (x, y) \in A^2_{\mu, \epsilon}$, we find $x^{**}_{\mu, \epsilon} \ge x, y^*_{\mu, \epsilon} \le y$. Thus, $(x^*_\mu, y^*_\mu)$ is the extreme point of $A^1_{\mu, \epsilon}$ and $(x^{**}_\mu, y^{**}_\mu)$ is the extreme point of $A^2_{\mu, \epsilon}$, we get the results. $\square$

## F   Experiments Details

In this section, we present experiments to validate the global convergence of Algorithm 6. Our goal is twofold: First, we aim to demonstrate that irrespective of the starting point, Algorithm 6 using gradient descent consistently returns the global minimum. Second, we contrast our updating scheme for $\mu_k, \epsilon_k$ as prescribed in Algorithm 6 with an arbitrary updating scheme for $\mu_k, \epsilon_k$. This comparison illustrates how inappropriate setting of parameters in gradient descent could lead to incorrect solutions.

## F.1 Random Initialization Converges to Global Optimum

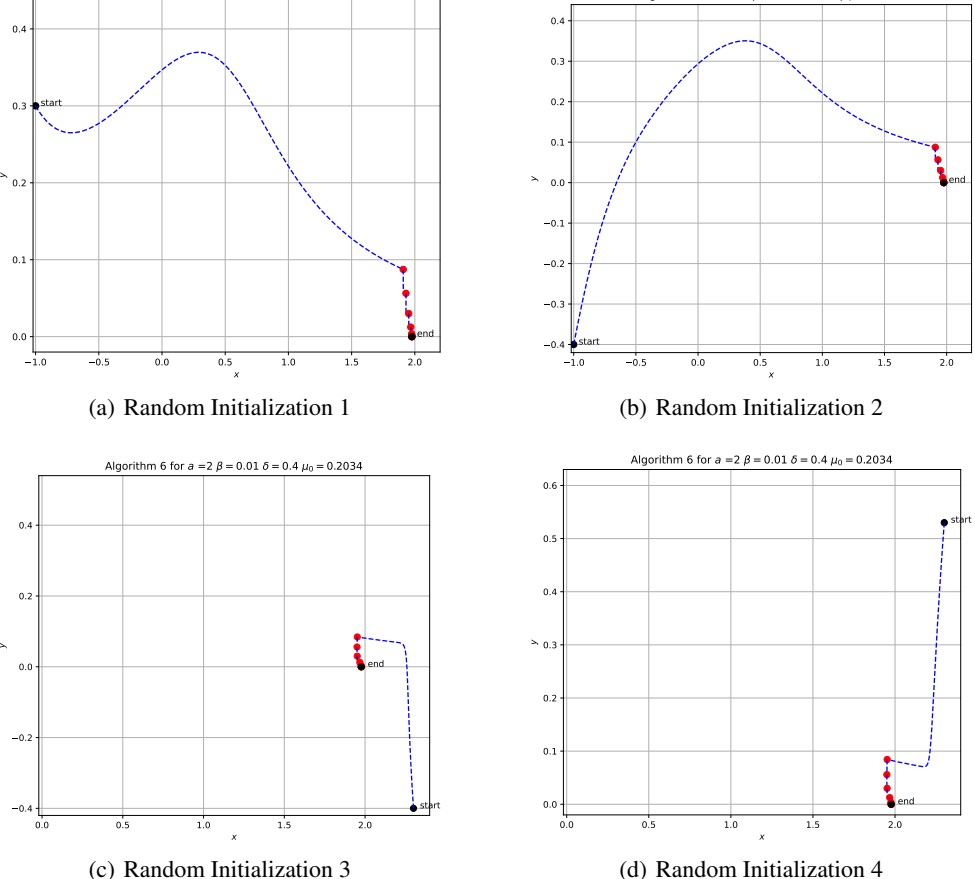

(a) Random Initialization 1

(b) Random Initialization 2

(c) Random Initialization 3

(d) Random Initialization 4

Figure 9: Trajectory of the gradient descent path with the different initializations for $a = 2$. We observe that regardless of the initialization, Algorithm 6 always converges to the global minimum. Initial $\mu_0 = \frac{a^2}{4} \frac{(1-\delta)^3 (1-\beta)^4}{(1+\beta)^2}$

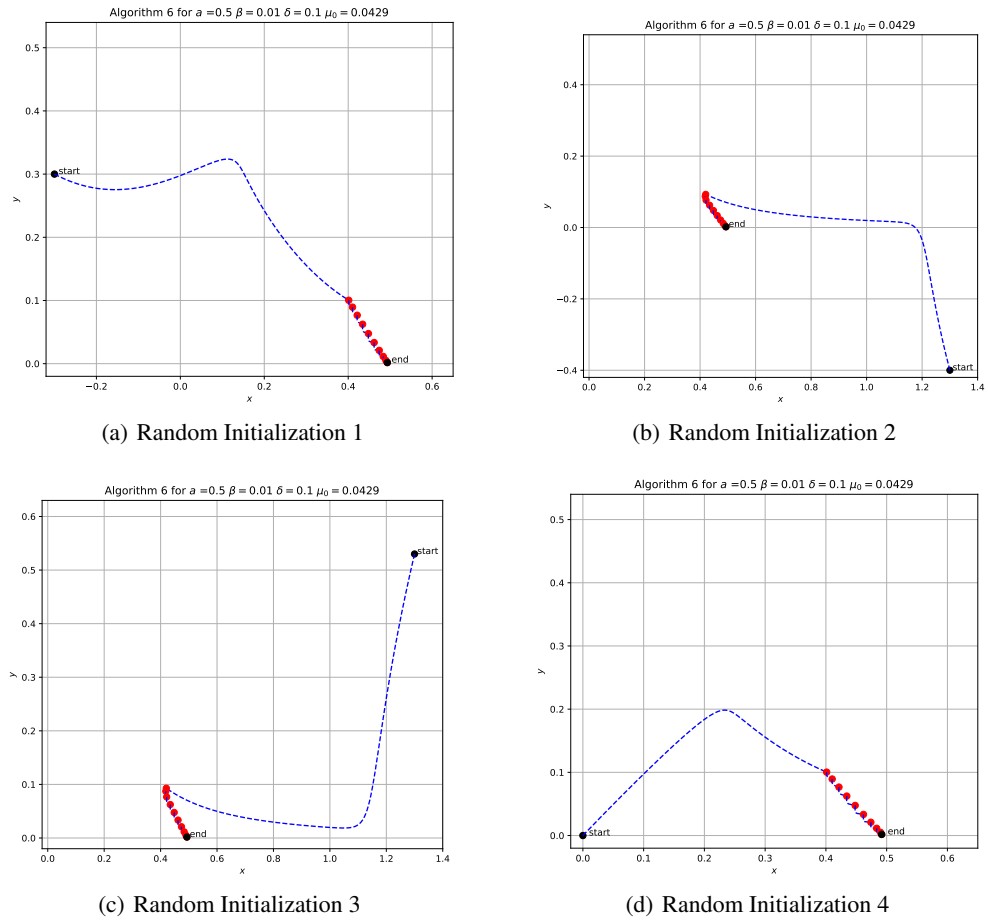

(a) Random Initialization 1

(b) Random Initialization 2

(c) Random Initialization 3

(d) Random Initialization 4

Figure 10: Trajectory of the gradient descent path with the different initializations for $a = 0.5$. We observe that regardless of the initialization, Algorithm 6 always converges to the global minimum. Initial $\mu_0 = \frac{a^2}{4} \frac{(1-\delta)^3(1-\beta)^4}{(1+\beta)^2}$

## F.2 Wrong Specification of $\delta$ Leads to Spurious Local Optimial

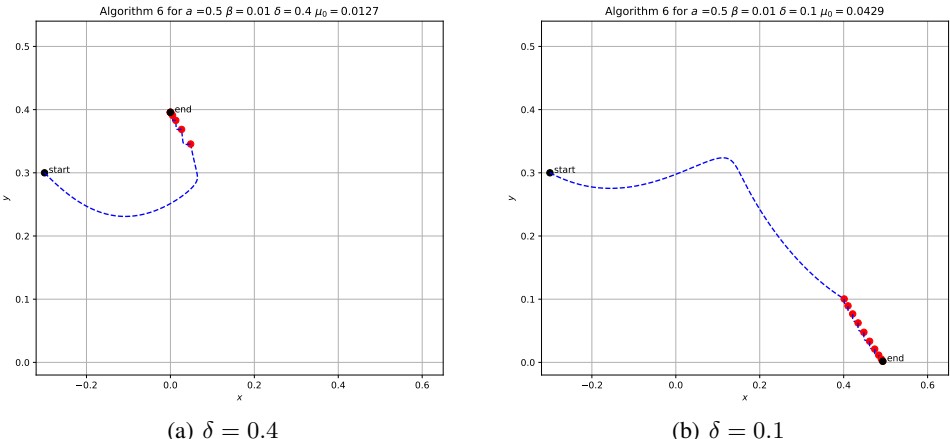

(a) $\delta = 0.4$

(b) $\delta = 0.1$

Figure 11: Trajectory of the gradient descent path for two difference $\delta$. Left: $\beta$ violates requirement $\left(\frac{1+\beta}{1-\beta}\right)^2 \leq (1-\delta)(a^2+1)$ in Theorem 4, leading to spurious local minimum. Right: $\beta$ follows requirement $\left(\frac{1+\beta}{1-\beta}\right)^2 \leq (1-\delta)(a^2+1)$ in Theorem 4, leading to global minimum. Initial $\mu_0 = \frac{a^2}{4}\frac{(1-\delta)^3(1-\beta)^4}{(1+\beta)^2}$

## F.3 Wrong Specification of $\beta$ Leads to Incorrect Solution

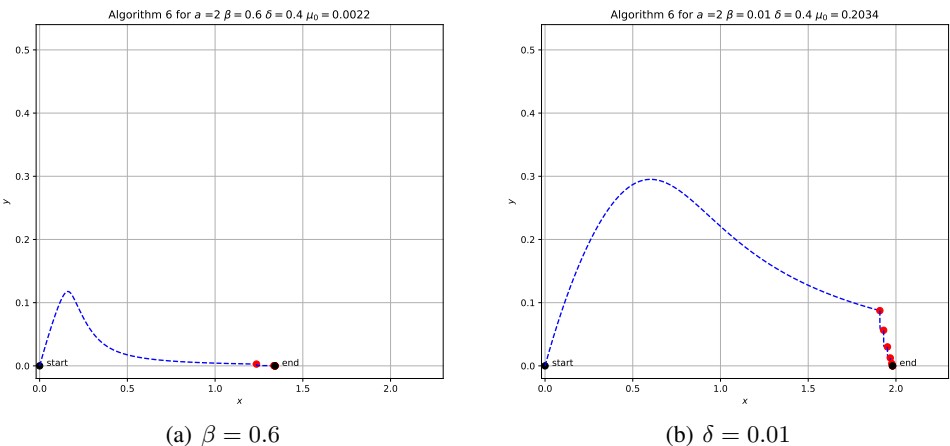

(a) $\beta = 0.6$

(b) $\delta = 0.01$

Figure 12: Trajectory of the gradient descent path for two difference $\beta$. Left: $\beta$ violates requirement $\left(\frac{1+\beta}{1-\beta}\right)^2 \leq (1-\delta)(a^2+1)$ in Theorem 4, leading to incorrect solution. Right: $\beta$ follows requirement $\left(\frac{1+\beta}{1-\beta}\right)^2 \leq (1-\delta)(a^2+1)$ in Theorem 4, leading to global minimum. Initial $\mu_0 = \frac{a^2}{4}\frac{(1-\delta)^3(1-\beta)^4}{(1+\beta)^2}$

## F.4   Faster decrease of $\mu_k$ Leads to Incorrect Solution

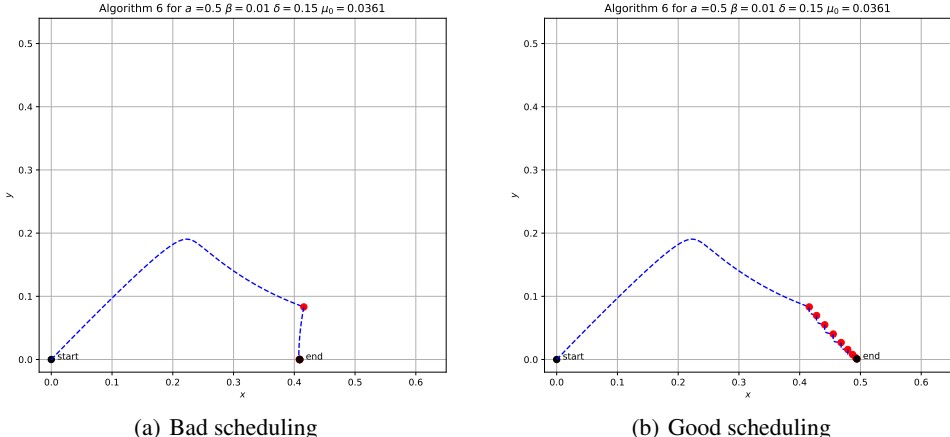

(a) Bad scheduling                     (b) Good scheduling

Figure 13: Trajectory of the gradient descent path for two difference update rules for $\mu_k$ with the same initialization. Left: "Bad scheduling" uses a faster-decreasing scheme for $\mu_k$, leading to an incorrect solution, even a non-local optimal solution. Right: "Good scheduling" follows updating rule for $\mu_k$ in Algorithm 6, leading to the global minimum. Initial $\mu_0 = \frac{a^2}{4} \frac{(1-\delta)^3 (1-\beta)^4}{(1+\beta)^2}$

