# OpenReview forum: "Global Optimality in Bivariate Gradient-based DAG Learning"
_NeurIPS.cc/2023/Conference — NeurIPS 2023 poster_

### Official Review · Reviewer_u4D3 · 2023-06-22

**Soundness:** 3 good
**Presentation:** 4 excellent
**Contribution:** 3 good
**Rating:** 8
**Confidence:** 3

**Summary:**

The authors give a simple optimization algorithm for DAG-learning-inspired optimization problems that avoids the limitations of known techniques.

**Strengths:**

Originality:
Work is original.
I particularly liked the reduction from a combinatorial problem to a non-convex optimization one.

Quality:
Simple and strong paper.

Clarity:
Clear writing, which can be improved; see questions.

Significance:
Significant topic/contributions.

**Weaknesses:**

No significant weaknesses found.

**Questions:**

Discussion around Equation (1) is very clean.

Line 49:
Can you please explain Equation (2) a bit more?

Line 64:
Is it really easy to see? :)

Line 90:
Please explain "homotopy."

Remark 2:
Can you please further explain the difficulties that you mention here?

Can you please elaborate on Equation (7)?

Line 217:
Can you please elaborate on bounding $a$?

Please add more details in the caption of Figure 2.

Line 235:
Why is the interesting regime the one where $\mu < \tau$?

Line 258:
Not clear!

Line 261:
"we know" instead of "we known."

**Limitations:**

Yes.

---

> ### Author Rebuttal · Authors · 2023-08-10
>
> We would like to express our gratitude to the reviewer for acknowledging the value of our contributions and our clear presentation.
>
> > Line 49: Can you please explain Equation (2) a bit more?
> >
>
> Equation (2) is a penalized version of Equation (1), where $h(W(\Theta))$ acts as a penalty. This change turns the initial constrained problem into a set of simpler unconstrained problems. The role of $\mu_k$ in Equation (2) is to control the magnitude of constraint violation by penalty. When $\mu_k$ is large, $f(\Theta)$ takes priority, so the solution minimizes this but may not meet the constraint $h(W(\Theta)) = 0$. By decreasing $\mu_k$, we give more weight to the penalty. As we approach $\mu_k = 0$, the solution focuses more on satisfying $h(W(\Theta))=0$.
>
> > Line 64: Is it really easy to see? :)
> >
>
> We will be sure to expand on this in the final version. Intuitively, let's recall that $f(\Theta)$ is presumed to be a convex function in our study. When $\mu_k$ is large, $f(\Theta)$ dominates in Equation (2). Consequently, $g_{\mu_k}(\Theta)$ behaves similarly to $f(\Theta)$ as a "convex" function, implying a benign loss landscape for $g_{\mu_k}(\Theta)$ when $\mu_k$ is large. This can be formally established as long as $f(\Theta)$ is a strongly convex function, which our population LS loss is.
>
> > Line 90: Please explain "homotopy."
> >
>
> “homotopy optimization", also known as "continuation optimization" [1, 2], is a strategy for finding solutions to complex optimization problems. The term "homotopy" is borrowed from topology (a branch of mathematics), which is a continuous transformation from one function to another. This concept is applied to optimization to form a bridge from a problem with a known or easy-to-find solution to a more complex problem. In our content, resolving $g_{\mu_k}(\Theta)$ becomes straightforward with larger values of  $\mu_k,$ while it becomes challenging when $\mu_k$ is small.
>
> > Remark 2: Can you please further explain the difficulties that you mention here?
> >
>
> The main challenge lies in the increasing complexity of the loss landscape of $g_{\mu}(\Theta)$ as the model dimensions increase. As a result, analyzing the basin of attraction for the global minimum of $g_{\mu}(\Theta)$ becomes increasingly difficult. Moreover, the penalty is an order-$d$ matrix polynomial, which becomes more complex as $d$ increases.
>
> > Can you please elaborate on Equation (7)?
> >
>
> Equation (7) is a detailed version of Equation (2). In the context of our study, we focus on the bivariate case, where the loss function is represented by the expected least square, given by $f(x,y) = \frac{1}{2}((1-ay)^2+y^2+(a-x)^2+1)$. The constraint function is denoted by $h(x,y) = \frac{x^2y^2}{2}.$
>
> Therefore, we can express $g_{\mu}(x,y) = \mu f(x,y)+h(x,y)=\frac{\mu}{2}((1-ay)^2+y^2+(a-x)^2+1)+\frac{x^2y^2}{2}$. The function $g_{\mu}(x,y)$ is used recurrently in our Algorithm 2 or 3, so our principal interest lies in understanding its properties.
>
> > Line 217: Can you please elaborate on bounding $a$?
> >
>
> Thanks for bringing it up! The lower bound on $a$ is indeed a standard and sensible assumption. One can view the magnitude of $a$ as a measure of problem difficulty, with $a$ essentially behaving like the “signal strength" of the underlying structure, larger $a$ indicates the structure is easier to learn. For a more direct insight, consider Equation (6)'s local optimal solution, namely $(x_1^*,y_1^*) = (0,\frac{a}{a^2+1})$. The corresponding loss is  $\frac{1}{2}(a^2+1+\frac{1}{a^2+1})$. Now, the global optimal solution is $(x_0^*,y_0^*) = (a,0)$, with loss $f(x^*_0,y^*_0) = 1$. The loss difference between the local and global optima is therefore $\frac{1}{2}(a^2+1+\frac{1}{a^2+1}) - 1$, an increasing function of $a$. As $a$ enlarges, both the loss difference and the basin of attraction for the global optimum increase in general, making the optimization easier.
>
> > Please add more details in the caption of Figure 2.
> >
>
> Thanks for the suggestion. Here, for $\mu>\tau,$ there exists a single solution to $r(y;\mu) = 0$, which implies there is one stationary point in Equation (7). When $\mu=\tau,$ two solutions are found for $r(y;\mu) = 0$, suggesting that there are two stationary points in Equation (7). Conversely, when $\mu<\tau,$ we observe three solutions for $r(y;\mu) = 0$, indicating that there are three stationary points in Equation (7)-  a local optimum, a saddle point, and a global optimum.
>
> > Line 235: Why is the interesting regime the one where $\mu<\tau$?
> >
>
> Thanks for bringing up this point! In this regime where multiple stationary points exist in $g_{\mu}(\Theta)$ - a local optimum, a saddle point, and a global optimum, as described in Lemma 1, the usage of gradient descent for $g_{\mu}(\Theta)$ may result in trap within the local optimal solution. This scenario is precisely what our methodology aims to avoid, our updating scheme of $\mu_k$ in Algorithm 2 or 3 can do the job. Conversely, when $\mu>\tau,$ only a single stationary point is observed. As demonstrated in Theorem 3 (Appendix), gradient descent invariably converges to this point, which consistently represents the global optimum. Hence, in such conditions, there are no concerns to be addressed.
>
> > Line 258: Not clear!
> >
>
> We are happy to explain this in a more accessible way. The fundamental idea is to identify the basin of attraction for the global optimum solution$(x^*_{\mu_{k+1}},y^*_{\mu_{k+1}})$ of $g_{\mu_{k+1}}(\Theta)$, and make sure previous initialization point $(x^*_{\mu_{k}},y^*_{\mu_{k}})$ fall into such region. We then use gradient descent or flow to converge towards $(x^*_{\mu_{k+1}},y^*_{\mu_{k+1}})$. This proof aligns with this reasoning and affirms that our updating scheme for $\mu_k$ can make it happen!
>
> [1] Lin, Yang, Zhang, and Zhang. "Continuation Path Learning for Homotopy Optimization." 2023.
>
> [2] Hazan, Levy, and Shalev-Shwartz. "On graduated optimization for stochastic non-convex problems." 2016.

---

### Official Review · Reviewer_Y11K · 2023-06-30

**Soundness:** 4 excellent
**Presentation:** 4 excellent
**Contribution:** 3 good
**Rating:** 7
**Confidence:** 3

**Summary:**

This paper studies the problem of learning the correct Directed Acyclic Graph (DAG) that describes the data using continuous optimization. The connection of this problem with continuous methods comes from a prior work of Zheng et al., where they introduce a differentiable function $h$ whose level set at 0 exactly characterizes DAGs. The problem then becomes to optimize a score function of the data subject to the constraint that h is 0. This is a non-convex problem, since h is non-convex. The standard way of solving it is by converting it to an unconstrained problem and penalizing the solutions with high values of h. This paper provides a way of solving a sequence of these problems with varying penalty parameters, so that the final output provably converges to the true solution, when we have two nodes in the DAG.  The model is analyzed in the population case, i.e. when we have access to infinite data.  The authors first show that for sufficiently small values of the penalty parameter, the loss landscape is not benign, as there exists a saddle point and a spurious local minimum in addition to the global minimum. Using this observation, they show a way to pick a sequence of penalty parameters, which provides a solution path that avoids the local minima and converges provably to the true model asymptotically. Their main result is phrased in terms of gradient flow, but they also provide a finite step analysis for gradient descent.

**Strengths:**

Consider the optimization problem
\begin{equation}\label{eq1}
\min_\Theta g_\mu(\Theta) := \mu f(\Theta) + h(W(\Theta))
\end{equation}
where $f$ is the least squares loss, $\mu$ is the penalty parameter, $\Theta$ represents the model parameters and $W(\Theta)$ represents the adjacency matrix of the DAG. $f$ is a function that was introduced in prior work, which is $0$ if and only if $W(\Theta)$ corresponds to a DAG.
This paper is the first that provides theoretical guarantees about the optimization landscape of this unconstrained problem for various values of $\mu$, in the case of $2$ nodes. It is worth noting that this is actually the method that is used in practice to learn DAGs, since enumeration methods quickly become intractable when the number of nodes in the DAG increases.
The authors establish interesting properties of the solutions space, namely that if $\mu$ is sufficiently small, the loss is not benign.
Also, the idea of successively solving different optimization problems and using the solution of the last as a starting point for the next is an intriguing one and could have applications beyond this work. The proof strategy is also very intuitive and elegant, by choosing $\mu$ so that the optimizer of $g_\mu$ lies in the basin of attraction of the global minimum of the next problem.
All arguments are very clearly explained, which makes this paper enjoyable to read.

**Weaknesses:**

-The result is only proven for $d=2$ nodes in the DAG. This makes most calculations tractable, since there is a closed form for the loss $g_\mu$ (a quadratic polynomial in two variables). Hence, while the proof strategy is elegant and novel, the technical arguments required to establish the claims do not have significant innovation.


**Questions:**

-It would be interesting to see whether similar results hold for higher values of $d$. Have the authors tried to run simulations for $d=3$ or $4$ and observe similar solution landscapes? What about other choices of loss function or penalty?
-Is it always true that there is a unique DAG that describes the data? For example, for two nodes, suppose we have $X_2 = X_1 + \epsilon$, where $\epsilon\sim \mathcal{N}(0,1)$. Then, clearly it is also true that $X_1 = X_2 + \epsilon'$, where $\epsilon' \sim \mathcal{N}(0,1)$. In that case, it seems that we have two solutions $x=1,y=0$ and $x=0,y=1$. I'm curious where would this come up in the analysis.

**Limitations:**

Yes

---

> ### Author Rebuttal · Authors · 2023-08-10
>
> We thank the reviewer for the time and effort into carefully reviewing our paper and provide such valuable feedback.
>
> > The result is only proven for $d = 2$ nodes in the DAG. This makes most calculations tractable, since there is a closed form for the loss  (a quadratic polynomial in two variables). Hence, while the proof strategy is elegant and novel, the technical arguments required to establish the claims do not have significant innovation.
> >
>
> We appreciate reviewer for acknowledging the analysis is elegant and novel! See discussion in **Common Concern II**.
>
> > It would be interesting to see whether similar results hold for higher values of $d$. Have the authors tried to run simulations for $d = 3$ or $d = 4$ and observe similar solution landscapes?
> >
>
> Thanks for the keen question! Please refer to **Common Concern I** for more details.
>
> > What about other choices of loss function or penalty?
> >
>
> We appreciate this interesting question!
>
> Crucially, the proof techniques in our work are notably general. Specifically, the implicit function theorem offers a means to trace the trajectory of stationary points, and the Lyapunov asymptotic stability theorem consistently can be used to identify the basin of attraction. These techniques clearly generalize to more general loss functions, which is an intriguing feature of our results. Extending our analysis in this way is an important future direction. Furthermore, it is worth emphasizing that under our model, the least squares loss is the most appropriate loss function: This is because its global minimizer is unique and equals the underlying matrix $W_*$.
>
> In relation to alternative penalties, we explored the matrix exponential $h_{\text{expm}}(W) = \operatorname{tr}(e^{W\circ W}) - d$  as described in [3] and the log-det formulation $h_{\text{ldet}}(W) = -\operatorname{logdet}(sI-W\circ W) +d\log s$ where $\rho(W\circ W)<s$(with $\rho$  representing the spectral radius) from [1]. Firstly, all these penalties are designed to enforce the DAG constraint on $W$. They are largely equivalent in application, implying that understanding one offers insights into the others. This equivalence is evident in experimental results; for instance, the loss landscapes are similar to each other. When $\mu$ falls below a certain threshold, the presence of local optima and saddle points for  $g_{\mu}(\Theta)$ remains consistent irrespective of whether $h_{\text{expm}}(W)$ or $h_{\text{ldet}}(W)$ is employed. However, incorporating $h_{\text{expm}}(W)$ or $h_{\text{ldet}}(W)$ introduces exponential or logarithmic terms to $g_{\mu}(W)$, adding layers of complexity to the analysis, as well as additional constraints in the case of $h_{\text{ldet}}$ since its validity rests on the condition $\rho(W\circ W)< s$.
>
> > Is it always true that there is a unique DAG that describes the data?
> >
>
> We appreciate your important question.  This is the identification problem in the DAG learning literature. In essence, the model is identifiable if different DAGs cannot generate the same distribution. In our setting, a linear model with equal (or known) error variances is identifiable (this is well-known, e.g. [5,6]). Therefore, solving Equation (6) will yield a unique DAG that accurately generates the data distribution. Although identifiability is not universal for all models, many have been confirmed as identifiable, e.g. [4,5,6,7,8]. Extending our techniques to these models is an important direction for future work. We hope this clarifies your question.
>
> > For example, for two nodes, suppose we have $X_2 = X_1 +\epsilon$, where $\epsilon \sim N(0,1)$. Then, clearly it is also true that $X_1 = X_2 +\epsilon'$ , where $\epsilon'\sim N(0,1)$. In that case, it seems that we have two solutions $x = 1,y = 0$  and $x = 0,y = 1$. I'm curious where would this come up in the analysis.
> >
>
> This is a great question! The issue boils down to independence of the noise terms (see L154), which is violated by your example (see below for details). Another way to interpret this assumption is that $N_2$ is independent of $X_1$, or more generally, that each noise term is independent of the parents in the structural equation model. See [5] or [6] for more discussion.
>
> Details: Your two models are (a)  $X_2 = X_1 +\epsilon$ and (b) $X_1 = X_2 +\epsilon'$ with $\epsilon'=-\epsilon$. Recalling our model assumptions that $(N_1,N_2)$ are independent and after transcribing the notation, in (a) this is equivalent to $X_1$ independent of $\epsilon$. Clearly, $X_2$ is not independent of $\epsilon$. But then in (b), $\epsilon'$ cannot be independent of $X_2$, which violates our model assumptions. So, there is only one model with independent noise terms. In fact, this argument generalizes quite substantially to LinGAM and additive noise models. In the Gaussian case there are some subtleties, but see [5-7] for details on this case.
>
> This fact does not explicitly come up in our analysis, but is implicit in the fact that only (a) will minimize the LS loss, a fact which follows from known results such as [5] or [6].
>
> [4] Peters, Jonas, Joris M. Mooij, Dominik Janzing, and Bernhard Schölkopf. "Causal discovery with continuous additive noise models." (2014).
>
> [5] Loh, Po-Ling, and Peter Bühlmann. "High-dimensional learning of linear causal networks via inverse covariance estimation." *The Journal of Machine Learning Research* 15, no. 1 (2014): 3065-3105.
>
> [6] Peters, Jonas, and Peter Bühlmann. "Identifiability of Gaussian structural equation models with equal error variances." *Biometrika* 101, no. 1 (2014): 219-228.
>
> [7] Aragam, Bryon, and Qing Zhou. "Concave penalized estimation of sparse Gaussian Bayesian networks." *The Journal of Machine Learning Research* 16, no. 1 (2015): 2273-2328.
>
> [8] Shimizu, Shohei, Patrik O. Hoyer, Aapo Hyvärinen, Antti Kerminen, and Michael Jordan. "A linear non-Gaussian acyclic model for causal discovery." *Journal of Machine Learning Research* 7, no. 10 (2006).

---

### Official Review · Reviewer_xJpx · 2023-07-15

**Soundness:** 2 fair
**Presentation:** 3 good
**Contribution:** 2 fair
**Rating:** 5
**Confidence:** 1

**Summary:**

This paper presents a novel approach to the non-convex optimization problems associated with learning the structure of a structural equation model (SEM) or Bayesian network. Considering the equivalent penalty form, the authors propose a homotopy-based optimization scheme that finds global minimizers of the problem by iteratively decreasing the penalty coefficient according to a given schedule. They prove that this algorithm converges globally to the global minimum, regardless of the initialization for W. The authors also demonstrate that the non-convex program is non-benign, meaning that naïve implementation of black-box solvers are likely to get trapped in a bad local minimum.

**Strengths:**


I am not an expert in this field. But the paper's findings seems to have significant implications in the context of learning the structure of SEMs or Bayesian networks.  The paper is well-structured and the authors clearly explain their methodology and findings. They also provide a clear visualization of the non-convex landscape and the solution trajectory.



**Weaknesses:**

The authors' approach is primarily focused on the bivariate case, which may limit its applicability in more complex settings. Some numerical study on large-scale problems is desirable.

**Questions:**



How does the landscape look like in high dimensional setting?  Can you provide any experiments to more than two variables?



**Limitations:**

Does not apply

---

> ### Author Rebuttal · Authors · 2023-08-10
>
> We sincerely thank the reviewer for the insightful critiques and comprehensive understanding of our work, and for providing such useful feedback. We will try our best to address the reviewer’s concern.
>
> > The authors' approach is primarily focused on the bivariate case, which may limit its applicability in more complex settings. Some numerical study on large-scale problems is desirable.
> >
>
> We appreciate the reviewer's insightful question about the scalability of the Homotopy method on large-scale problems. Indeed, this is exactly what is done in practice on large-scale problems (see discussion in Line 84-87). Since this point is closely related to your follow-up questions, please see below for more details.
>
> > Can you provide any experiments to more than two variables?
> >
>
> Many existing papers show that the Homotopy method can perform well empirically, even with hundreds or thousands of nodes.  For example, the study presented in [1] employs the Homotopy method to solve Equation (1) and has achieved state-of-the-art results, as demonstrated in Figures 4, 5, and 6 in [1], and further elaborated in their appendix. The experiments, encompassing both linear and nonlinear models, offer compelling evidence in support of the homotopy method's effectiveness in practice. Furthermore, while [2] and [3] don't adopt the Homotopy algorithm in its exact form, their approaches share a similar spirit of the homotopy algorithm, solving Equation (2) repeatedly with previous solutions. For more empirical results, see Figure 1 and Appendix H in [2], and Figure 3,7,8 in [3]. More discussions can be found in **Common Concern II**.
>
> > How does the landscape look like in high dimensional setting?
> >
>
> Thanks for this insightful question. In higher-dimensional settings, the landscape remains non-benign (e.g. even for $d = 3$, experiments indicate that multiple local optima and saddle points persist), however, existing work [1,3] has shown that the homotopy method is still very effective at finding good (in some cases, near optimal) solutions. This is in fact part of the motivation for our study. Extending our results to higher dimensions is an important direction for future work, and it is worth pointing out that our techniques are indeed generalizable in principle: The main tools we use (the implicit function theorem and the Lyapunov asymptotic stability theorem) possess broad applicability, extending beyond two dimensions. While the direct translation of our findings to other contexts remains challenging, they undeniably pave the way for future explorations. More discussions can be found in **Common Concern I**.

---

> > ### Comment · Reviewer_xJpx · 2023-08-19
> >
> > Thank you for the clarification. There could be some interesting future directions to pursue, but my concern about the limitation of this current paper is not fully addressed. I will keep my score.

---

### Official Review · Reviewer_NK3q · 2023-07-31

**Soundness:** 3 good
**Presentation:** 2 fair
**Contribution:** 2 fair
**Rating:** 5
**Confidence:** 3

**Summary:**

the paper provides a theoretical study on the loss landscape and convergence in gradient-based DAG learning framework. By focusing on linear functions with the number of variable d = 2, They provide a homotopy-based optimization scheme to guarantee the global optimality. Some numerical validations are provided.

**Strengths:**

the paper focuses on the theoretical study on the global optimality and convergence. This is an important but difficult questions. The theoretical results mainly shows the initialization regime as requirements for convergence, which makes sense given the nonconvex nature, although I did not check the proof in details (given its length).

Illustrative examples and results are provided.

**Weaknesses:**

Unfortunately, the current results are only applicable to d=2 and linear functions (understandably, of course). No discussions on how nonlinear or  a large number of variables would affect the loss landscape, and/or how the homotopy algorithm would be affected.

Would be more interesting if the insights of the theoretical results can be used in practical algorithms.



**Questions:**

- "...bears a resemblance to the success of training deep models, which started with AlexNet for image classification": First of all, modern deep models have success before alexnet on imagenet, so this is not accurate. Second, such a statement is over the top and should be removed.
- L82: indeed only two DAGs are in the entire search space yet the analysis is quite complex. There is no indication on how such analysis and approach would scale or behave with an increasing number of variables, hence it is hard to judge the usefulness of these analysis yet. In comparison, although the combinatorial search is not used, its expected performance is the same with larger d.
- Eq 4: x is already used to represent data, and should avoid using it as parameter (lack of notation clarity).
- Eq 5: while re-ordering variable to obtain such a upper triangular structure is not an issue here yet, in datasets W* would generally not be so. How does a non-upper triangular structure impact analysis, for example a's usage afterward?
- Some derivation are not given fully. For example, f(W) and h(W) skips many steps in between.
- L199: what is k here?
- the regime of initialization for \mu: very interesting observation.
- optimization landscape with nonlinear functions: current analysis focuses on linear function. How would it change for nonlinear functions (such neural networks), even for d=2? Given the nonlinear nature, one can imagine it would make the analysis even hard, given two layers of neural networks are the most complex cases that has some theoretical understanding.
- can the homotopy-based optimization scheme be used to improve convergence of NOTEARS related algorithms?

**Limitations:**

not discussed but not needed.

---

> ### Author Rebuttal · Authors · 2023-08-10
>
> We would like to express our gratitude to the reviewer for their time, effort, and valuable suggestions.
>
> > No discussions on how nonlinear or a large number of variables would affect the loss landscape, and/or how the homotopy algorithm would be affected.
> >
>
> Thanks for this insightful question! See **Common Concern I**.
>
> > Would be more interesting … be used in practical algorithms.
> >
>
> It’s excellent point!  We go deeper into this in the discussion below.
>
> > can the homotopy-based optimization scheme be used to improve convergence of NOTEARS related algorithms?
> >
>
> Indeed, our homotopy-based scheme is proposed precisely because it is what is used in practice [1][4]! Thus, our theory provides guarantee that homotopy methods are indeed a viable approach in practice. Concretely, our results show that tuning the penalty schedule is important, and that global initialization is possible if the signal strength is sufficiently large. When done properly, convergence is very fast (exponential).
>
> In more detail: When considering both Algorithm 2 and Algorithm 3, any given initialization can converge towards the global optimum, consequently, there's no need to prioritize specific initializations. Furthermore, note that any initial penalty parameter, $\mu_0$, fall into a specific range can achieve this, negating the need to set $\mu_0$ to an exceedingly large value to ensure a favorable landscape for $g_{\mu}(\Theta)$(i.e., one that only possesses a singular stationary point), as a result, reducing the number of outer iterations in Algorithms 2 and 3. As for the decay rate of $\mu_k$, it can be quite rapid -  in specific terms, an exponential decay. Our analysis offers clarity on the explicit dependence between convergence rates and various parameters, e.g. Theorem 2 in the paper and Theorem 4 in appendix for details. In principle, this knowledge can be used to accelerate NOTEARS-related algorithms, which is an important direction for future work. To illustrate, a greater weight in $W_*$ can allow for a more progressive decay of $\mu_k$ and accommodate a smaller initialization for $\mu_0$, as a result, faster convergence. Empirically, these theoretical conclusions are aligned with our experimental findings (arbitrary initialization, and convergence in a few iterations.) The experiments detailed in [1] confirm this – they begin with a zero initialization point and achieve their objectives in a mere four iterations, utilizing $\mu$ values set at $\{1,0.1,0.001,0\}$, even for hundreds and thousands of nodes.
>
> > "...bears a resemblance … over the top and should be removed.
> >
>
> Thanks for flagging this. in hindsight we see how this is a bit over the top. We will take out this comment in the final version.
>
> > L82: indeed only two DAGs … the same with larger d.
> >
>
> We appreciate the reviewer's insightful question. See **Common Concern II** for more details.
>
> > Eq 5: while re-ordering … for example a's usage afterward?
> >
>
> Thank you for the insightful observation! Absolutely, in a broader context, $W_*$ could be neither an upper nor a lower triangular matrix, posing a considerable challenge for analysis. Determining how this affects our analysis is an interesting future direction. Nonetheless, there is another finding that could be useful. The underlying structure decides the nature of the analysis. For instance, for a three-node system with variables *$X_1, X_2, X_3$,* the analysis for a chain structure like $X_1\rightarrow X_2\rightarrow X_3$ should mirror that of $X_2\rightarrow X_1\rightarrow X_3$, given their similar structure in essence. Conversely, the analysis would be different if we compare the former chain with a collider structure like $X_1\rightarrow X_2\leftarrow X_3$. Yet, when we constrain the dimensions to $d=2$, the two choices are $X_1\rightarrow X_2$ or $X_2\rightarrow X_1$, thus, $W_*$ will either be an upper or lower triangular matrix. Consequently, the scenarios become equivalent, suggesting that our analysis for one suffices for both.
>
> > the regime of initialization for $\mu$: very interesting observation.
> >
>
> Thanks for acknowledging this interesting observation. This is based on our analysis of $g_{\mu}(\Theta)$, and such regime yields several compelling implications. First, as $a$ increases, the regime typically expands, suggesting the problem get easier when the underlying structure has stronger signal (e.g. $a$ in Equation (5)). Second, any initial penalty parameter, $\mu_0$, fall into a specific range can achieve this, negating the need to set $\mu_0$ to an exceedingly large value to ensure a favorable landscape for $g_{\mu}(\Theta)$. Finally, it is essential to notice that such regime is the key for us to figure out an universal initialization $\mu_0 = \frac{1}{27}$ for Algorithm 3, since $\frac{a^2}{4(a^2+1)^3}$ is upper bounded by $\frac{1}{27}$.
>
> > optimization landscape with nonlinear functions… theoretical understanding.
> >
>
> This is a very good question! When neural networks are introduced into the mix, the loss function $g_{\mu}(\Theta) = \mu f(\Theta )+h(W(\Theta))$ inherits a dual-layer complexity. The first layer arises from the inherent nonconvexity of the neural network as seen in the loss $f(\Theta)$, while the second stems from the nonconvexity of the DAG constraint present in $h(W(\Theta))$. Such compounded intricacies can be expected to substantially alter the landscape of $g_{\mu}(\Theta)$, potentially introducing numerous local optima and saddle points. Although we currently are uncertain on this matter, it's an interesting direction to explore how advancements in the study of two-layer neural networks can be integrated with the techniques we employed, such as the Lyapunov asymptotic stability theorem and the implicit function theorem, to better understand the theoretical facets of DAG learning using neural networks.
>
> [4] Ng, Lachapelle, Ke, Julien, and Zhang. "On the convergence of continuous constrained optimization for structure learning." 2022.

---

### Author Rebuttal · Authors · 2023-08-10

To all reviewers,

We thank all reviewers for their time put into reading our work and their valuable comments. We appreciate the consensus that our paper is well-written and theoretical contributions are delivered clearly. Finally, we appreciate Reviewer Y11K acknowledging our analysis elegant and novel. We next respond to the common concerns.

*Due to character limits, any minor concerns omitted will certainly be addressed based on the reviewers' suggestions in the final version.*

*Regarding the other major concerns raised, we will respond to each reviewer individually.*


**Common Concern I**

> How landscape is affected for high dimensional setting and how the homotopy algorithm would be affected.
>

This is a very insightful question. In higher-dimensional settings, the landscape remains challenging, however, previous work has convincingly demonstrated the utility and applicability of homotopy-based approaches in practice. Indeed, the original NOTEARS implementation as well as more recent advances use some form of homotopy algorithm. See [1,2,3] for details. Although homotopy methods generally faces challenges with getting trapped in local optima, there are certain scenarios where this can be potentially circumvented. Based on our own experiments, global convergence remains feasible in these instances:

- Increasing the signal strength (e.g. $a$ in Equation (5)) appears to prevent the Homotopy method from being trapped into local optima. This weight can be seen as the signal strength of the underlying structure, and enhancing it could also alleviate the "trapping into local optimum" issue. This aligns with our findings in the case of $d = 2$; when $a > \sqrt{5/27}$, there exists a universal updating scheme for $\mu_k$ that functions effectively, and global convergence is guaranteed as stated in Corollary 1.
- Enhancing the sparsity of the underlying structure indeed seems to mitigate the risk of the Homotopy method getting trapped into local optima. This intriguing observation coincides with prior experimental results in [1][2][3], which indicate that sparse graphs are generally easier to learn. As such, this important discovery further reinforces the value of investigating sparsity as a potential direction for future work.

In higher-dimensional settings, the landscape remains non-benign. For instance, even when $d = 3$, experiments indicate the presence of multiple local optima and saddle points. Determining how to incorporate sparsity structure and signal strength into the analysis to ensure global convergence of the homotopy method is an important direction for future work.

**Common Concern II**

> How such analysis and approach would scale or behave with an increasing number of variables?
>

This is really insightful point!

- Firstly, many existing papers suggest that the Homotopy method can perform well empirically, even with hundreds or thousands of nodes. This is described in our paper (specifically, Remark 1, L84-87). The study presented in [1] employs the Homotopy method to solve Equation (1) and has achieved state-of-the-art results, as demonstrated in Figures 4, 5, and 6 in [1], and further elaborated in their appendix. Furthermore, while [2] and [3] don't adopt the Homotopy algorithm in its exact form, their approaches share the similar spirit of homotopy algorithm, solving Equation (2) repeatedly with previous solutions. It is essential to note, however, that a theoretical understanding of the convergence dynamics and the loss landscape of $g_{\mu}(\Theta)$ remains largely unexplored. Our investigation into the bivariate scenario aspires to address this gap, offering a theoretical framework to explain and improve these known empirical findings.
- Secondly, we would like to emphasize that the current analysis is intuitive but far from trivial and has the potential to inform more complex settings. Initially, we delineate the conditions leading to multiple stationary points of $g_{\mu}(\Theta)$ as evidenced by Theorems 5, 6(v) – a process that relies heavily on a clever analysis of two related objectives $r(y_{\text{ub}};\mu)$ and $t(x_{\text{lb}};\mu)$ , especially given the absence of explicit formula for the solutions to $r(y;\mu) = 0$  and $t(x;\mu) = 0$ (see proof of Theorems 5,6 in appendix). Subsequently, we leverage the implicit function theorem to identify the evolving trajectories of these stationary points as a function of $\mu$ (see Theorem 5, 6(vi)), pinpointing the precise trajectory to pursue. Finally, our use of the Lyapunov asymptotic stability theorem allows us to define the basin of attraction for the correct stationary point, detailed in the proof of Lemma 7. Each of these components not only presents an intuitive understanding but can also serve as foundational blocks for more intricate cases. It's noteworthy that tools like the implicit function theorem and the Lyapunov asymptotic stability theorem possess broad applicability, extending beyond the confines of linear models. While the direct translation of our findings to other contexts remains challenging, they undeniably pave the way for future explorations.

[1] Bello, Kevin, Bryon Aragam, and Pradeep Ravikumar. "Dagma: Learning dags via m-matrices and a log-determinant acyclicity characterization." *Advances in Neural Information Processing Systems* 35 (2022): 8226-8239.

[2] Ng, Ignavier, AmirEmad Ghassami, and Kun Zhang. "On the role of sparsity and dag constraints for learning linear dags." *Advances in Neural Information Processing Systems* 33 (2020): 17943-17954.

[3] Zheng, Xun, Bryon Aragam, Pradeep K. Ravikumar, and Eric P. Xing. "Dags with no tears: Continuous optimization for structure learning." *Advances in neural information processing systems* 31 (2018).

---

### Decision · Program_Chairs · 2023-09-21

**Decision:**

Accept (poster)

**Comment:**

This work addresses the DAG learning problem. Despite being a nonconvex optimization problem with the possibility of multiple spurious solutions, the authors claim that their proposed algorithm can find the global optimal solution for the population loss but only in the bivariate setting. However, it's important to note that the theoretical result is limited to cases where $d=2$, which is not practical in most real-world scenarios.

All reviewers have generally provided positive feedback. While the work is interesting and may have some scientific impact, it currently lacks practical usability.